# Model-free estimation of completeness, uncertainties, and outliers in atomistic machine learning using information theory

Daniel Schwalbe-Koda [1,2] ✉, Sebastien Hamel [1], Babak Sadigh[1], Fei Zhou [1] & Vincenzo Lordi [1] ✉

An accurate description of information is relevant for a range of problems in atomistic machine learning (ML), such as crafting training sets, performing uncertainty quantification (UQ), or extracting physical insights from large datasets. However, atomistic ML often relies on unsupervised learning or model predictions to analyze information contents from simulation or training data. Here, we introduce a theoretical framework that provides a rigorous, model-free tool to quantify information contents in atomistic simulations. We demonstrate that the information entropy of a distribution of atom-centered environments explains known heuristics in ML potential developments, from training set sizes to dataset optimality. Using this tool, we propose a model-free UQ method that reliably predicts epistemic uncertainty and detects out-of-distribution samples, including rare events in systems such as nucleation. This method provides a general tool for data-driven atomistic modeling and combines efforts in ML, simulations, and physical explainability.

Quantifying information contents in datasets is an essential task in many fields of science. Information theory, initially proposed in the context of communication theory[1], enabled a rigorous treatment of data, errors, and information in fields beyond its own, such as statistical thermodynamics[2], biophysics[3], or deep learning[4]. Particularly in the materials and chemical sciences, the known parallels between thermodynamic entropy and information theory[1] have long inspired quantitative descriptions of information contents in atomistic data. Simulation outcomes have been extensively related to thermodynamic entropy[5–11], often with an explicit definition of the degrees of freedom for the systems. Nevertheless, the need for an information metric in atomistic data goes beyond that from statistical thermodynamics, especially within modern computational materials science. For instance, machine learning interatomic potentials (MLIPs) have showed great promise in bypassing density functional theory (DFT) in atomistic simulations[12–22], but their development requires careful dataset construction, reliable uncertainty quantification (UQ) strategies, and statistical tools to assess the reliability of production simulations to reflect the chosen

(e.g., DFT) ground truth (Fig. 1a). At some level, these tasks require assessing how much information is present in the input or output data. These requirements are particularly critical in the case of neural network (NN)-based MLIPs, whose costly training processes and unreliable extrapolation performances require efficient training techniques and improved reliability in predictions[23–25]. Furthermore, detection of rare events, order parameters, and accessible phase spaces is an important task within atomistic simulations, but often requires hand-crafted descriptors. Thus, understanding information contents within atomistic data and rigorously quantifying them is essential for improving training efficiency, robustness, and interpretability of machine learning (ML)-driven simulations.

In this work, we propose a method to quantify information contents in atomistic simulation data, and demonstrate its application in a range of tasks related to atomistic simulations and MLIPs. Specifically, building on the mathematical formalism of information theory, we show that the information entropy from a distribution of atom-centered representations can be used to: (1) explain trends in MLIP

[1]Lawrence Livermore National Laboratory, Livermore, CA 94550, USA. [2]Department of Materials Science and Engineering, University of California, Los Angeles, CA 90095, USA. ✉e-mail: dskoda@ucla.edu; lordi2@llnl.gov

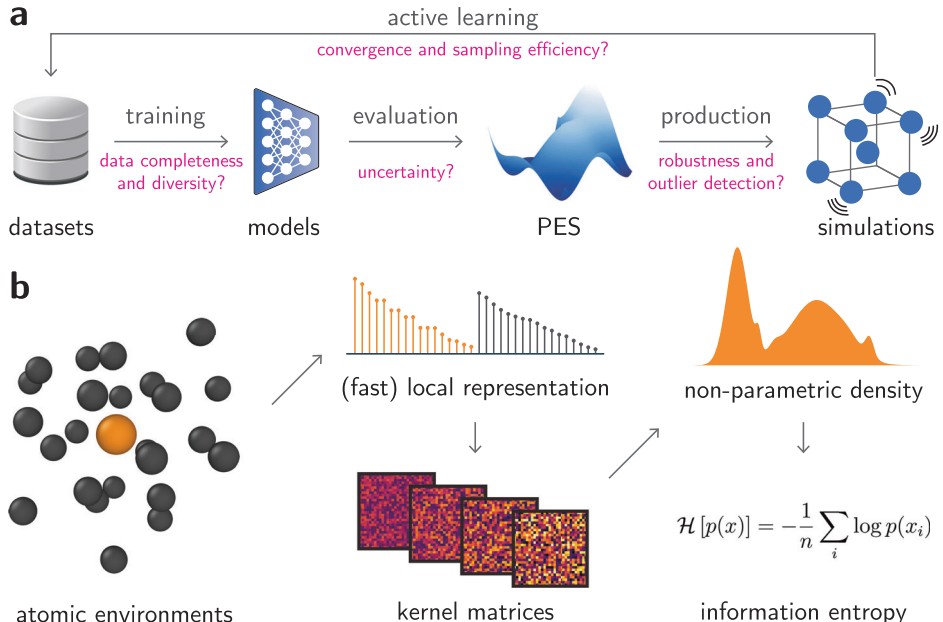

**Fig. 1 | Overview of the method. a** Typical workflow in machine learning interatomic potentials for training, evaluating, and retraining models that predict potential energy surfaces (PES). Challenges in the process are highlighted in magenta. **b** Overview of our method, Quick Uncertainty and Entropy from STructural Similarity (QUESTS), which computes the information entropy $\mathcal{H}$ of a non-parametric descriptor distribution $p(x)$ containing $n$ samples $x_i$.

errors even in the absence of model predictions; (2) rationalize dataset analysis, enabling the quantification of diversity, data efficiency, and convergence in active learning loops; (3) provide a model-free uncertainty estimate for ML-driven simulations; and (4) detect outliers in atomistic simulations, which can be used to identify failures in ML-driven simulations or even rare events such as the onset of nucleation. Importantly, this work goes beyond qualitative parallels with information theory and develops an approach where information contents can be quantified under the assumptions of that theory. This allows us to use known results in information theory to enable data compressibility, efficiency, error detection, and many others, in the context of atomistic simulations. This work provides a toolkit for atomistic simulations, MLIP development, and UQ for computational materials science, and can be extended to enable faster and more accurate materials modeling beyond predictions of potential energy surfaces (PES).

## Results

### Formulation of an atomistic information entropy

To approximate a one-to-one mapping between atomistic environments and the probability distributions from which the data is sampled, we propose a descriptor for atomic environments inspired by recent studies in continuous and bijective representations of crystalline structures[26] and similar to the DeepMD descriptors[17]. Given their success in a range of applications[17,27], the representation offers a rich metric space without sacrificing its computational efficiency, allowing us to perform non-parametric estimates of data distributions even in large datasets ( >$10^6$ atoms). Despite its simplicity, it also mitigates discontinuities in the representation space within controlled hyperparameter settings (see Supplementary Text). To obtain the descriptor, we begin by sorting the distances from a central atom $i$ to its $k$-nearest neighbors (within periodic boundary conditions, if appropriate) and obtain a vector $\mathbf{X}_i^{(1)}$ with length $k$,

$$\mathbf{X}_i^{(1)} = \left[ \frac{w(r_{i1})}{r_{i1}} \quad \cdots \quad \frac{w(r_{ik})}{r_{ik}} \right]^T, \quad r_{ij} \le r_{i(j+1)}, \tag{1}$$

with $1 \le j \le k$ due to the $k$-nearest neighbors approach and $w$ a smooth cutoff function given by

$$w(r) = \begin{cases} \left[ 1 - \left( \frac{r}{r_c} \right)^2 \right]^2, & 0 \le r \le r_c, \\ 0, & r > r_c. \end{cases} \tag{2}$$

However, the radial distances alone do not capture bond angles and cannot be used to fully reconstruct the environment. To address this issue, we construct a second vector to augment $\mathbf{X}_i^{(1)}$ that aggregates distances from neighboring atoms, inspired by the Weisfeiler-Lehman isomorphism test and analogous to message-passing schemes in graph neural networks (see Methods and Fig. S2):

$$X_{in}^{(2)} = \left\langle \frac{\sqrt{w(r_{ij})w(r_{il})}}{r_{jl}} \right\rangle_n, j, l \in \mathcal{N}(i), X_{in} \ge X_{i(n+1)}, \tag{3}$$

where $j$ and $l$ are atoms in the neighborhood $\mathcal{N}$ of atom $i$, $\langle . \rangle_n$ represents the average of the $n$-th elements of the sequence (details in the Supplementary Text), and $1 \le n \le k - 1$ due to the number of $k$-nearest neighbors pairs. The final descriptor $\mathbf{X}_i$ is obtained by concatenating $\mathbf{X}_i^{(1)}$ and $\mathbf{X}_i^{(2)}$. As this representation requires only the computation of a neighbor list, it can be easily parallelized and scaled to large systems.

To quantify the information entropy from a distribution of feature vectors $\{\mathbf{X}\}$, we start from the definition of the Shannon entropy $\mathcal{H}$[1],

$$\mathcal{H}[p(x)] = -\int p(x) \log p(x) dx, \tag{4}$$

where log is the natural logarithm, implying that $\mathcal{H}$ is measured in units of nats. Given a kernel $K_h$ with bandwidth $h$, we can perform a kernel density estimate (KDE) of a distribution of $n$ atomic environments $\mathbf{X}_i$ to obtain a non-parametric estimate of the information entropy of $p(x)$[28],

$$\mathcal{H}(\{\mathbf{X}\}) = -\frac{1}{n} \sum_{i=1}^{n} \log \left[ \frac{1}{n} \sum_{j=1}^{n} K_h(\mathbf{X}_i, \mathbf{X}_j) \right], \tag{5}$$

which corresponds to a discrete version of the original information entropy in Eq. (4) (see derivation in the Supplementary Text). If the kernel is defined in the space $K_h : \mathbb{R}^N \times \mathbb{R}^N \to [0, 1]$, then the entropy from Eq. (5) recovers useful properties from information theory such as well-defined bounds ($0 \le \mathcal{H} \le \log n$) and quantifies the absolute amount of information in a dataset {**X**}. This agreement with information-theoretical definitions contrasts with other relative metrics of entropy in atomistic datasets such as the ones from Perez et al.[29] or Oganov and Valle[30], which can be ill-defined in the presence of identical feature vectors or structural outliers. In our definition, $\mathcal{H} = \log n$ implies $K_h(\mathbf{X}_i, \mathbf{X}_j) = \delta_{ij}$, which is the case when all points are dissimilar from each other. $\mathcal{H} = 0$, on the other hand, implies $K_h(\mathbf{X}_i, \mathbf{X}_j) = 1, \forall i, j$, which represents a degenerate dataset with all points equivalent to each other. This work explores how this and other properties of Eq. (5) are useful for a variety of applications in atomistic simulations, specifically in the quantification of errors, uncertainties, and outliers in model-free regimes. A more complete discussion for the entropy estimation in atomistic datasets, especially the properties particular to this mathematical formalism, are described in detail in the Supplementary Text.

To employ Eq. (5) in practice, we choose $K_h$ to be a Gaussian kernel,

$$K_h(\mathbf{X}_i, \mathbf{X}_j) = \exp\left(\frac{-\left\| \mathbf{X}_i - \mathbf{X}_j \right\|^2}{2h^2}\right), \qquad (6)$$

where the bandwidth $h$ is selected to rescale the metric space of **X** according to the distance between two FCC environments with a 1% strain (Supplementary Text, Sec. A.6). Nevertheless, as the choice of kernel is known to influence the estimated distribution, the bandwidth (and associated entropy) may vary according to the kernel and may have to be calibrated depending on the dataset. Within this work, we show that even a constant bandwidth was found to be quite adequate for a range of datasets and tasks adopting this Gaussian kernel.

To quantify the contribution of a data point **Y** to the total entropy of the system, we define the differential entropy $\delta\mathcal{H}$ as

$$\delta\mathcal{H}(\mathbf{Y}|\{\mathbf{X}_i\}) = -\log\left[\sum_{i=1}^{n} K_h(\mathbf{Y}, \mathbf{X}_i)\right], \qquad (7)$$

where $\delta\mathcal{H}$ is defined with respect to a reference set {**X**} and can assume any real value (Supplementary Text, Sec. A.4). The measure $\delta\mathcal{H}$ intuitively represents how much "surprise" there is in a new point **Y** given the existing observations {$\mathbf{X}_i$}, and will be shown to enable uncertainty quantification, outlier detection, and other important results.

An overview of this method, named Quick Uncertainty and Entropy from STructural Similarity (QUESTS), is shown in Fig. 1b, and a range of examples and visualizations demonstrating the intuition behind the method are provided in the Supplementary Text (Figs. S1–S9). The code is available at https://github.com/dskoda/quests.

## Information-theoretical dataset analysis for machine learning potentials

Most recent MLIPs predict potential energy surfaces from fixed or learned atom-centered representations, similar to the strategy adopted in this work. Despite the wide usage of these models, constructing training datasets for these potentials is still a challenge[31–34]. Works such as the ones from Perez et al. proposed quantifying entropy as a way to build diverse atomistic datasets[29,33], but their approximation to entropy in the descriptor space prevents recovering true values of information from datasets, as defined by information theory. Furthermore, while using large amounts of data to train models can enhance the generalization power of NNIPs[35–37], naïvely generating

large, redundant datasets can be a counterproductive strategy; rather, training costs may be reduced by creating smaller optimal datasets that still achieve similar or even better results[34]. Since generating training data generally involves computationally expensive ground-truth calculations and training a model on a larger dataset leads to increased training cost and higher training complexity, there is significant advantage in understanding how to simultaneously minimize dataset size, maximize their coverage in the configuration space, and still maintain the accuracy of the MLIP trained on the full dataset.

**Relating information contents to learning curves in molecular datasets.** Borrowing from a fundamental concept in information theory, we hypothesize that the information entropy of atomistic datasets relates to the limit of their (lossless) compression and can thus explain results from learning curves in MLIPs. The theoretical results from information theory already guarantee the compression limits that can be applied to any data[1], but it is not clear whether the same effect can be observed in atomistic datasets. If true, this enables us to: (1) explain trends in learning curves in ML potentials; (2) quantify redundancy in existing datasets; and (3) evaluate the sampling efficiency of iterative dataset generation methods (Fig. 1a). As an initial test for (1), we computed the entropy $\mathcal{H}$ as a function of dataset size of different molecules in the rMD17 dataset[16,38], which has been widely used to evaluate the performance of different MLIPs. The bandwidth was set to a constant value of 0.015 Å$^{-1}$ to ensure that data points have small overlap, which represents an underestimation of the extrapolation power of MLIPs (see Methods for more details on the choice of bandwidth parameter). The information entropy of three examplar molecules is shown in Fig. 2a (Fig. S10 for all molecules). In the low data regime, the total dataset entropy increases rapidly with the number of samples. On the other hand, in the high data regime, the values of $\mathcal{H}$ saturate because little novelty is obtained from more data points sampled from the same MD trajectories. As expected, the saturation point depends on the molecule under analysis. Benzene, a stiff molecule with six redundant environments for both carbon and hydrogen, reaches its maximum entropy in less than 100 samples. Azobenzene, a molecule with atomic environments exhibiting two- and four-fold degenerate environments, is nearly at its maximum entropy value at 1000 samples. Although our method does not consider element types, elements may sometimes be distinguished by their environments (i.e., valence rules) and diversity of vibrational motion, allowing the (approximate) quantification of information entropy in a trajectory. The information on the MD trajectory of aspirin, a much more diverse molecule, is not fully converged even at 10,000 samples. As this molecule has more rotatable bonds and unique atomic environments than its counterparts, it is expected that its information content is larger, as shown by its higher entropy, and requires more samples to saturate.

We hypothesize that the mismatch between the amount of information in each molecule and the constant number of samples used can partially explain the trends in testing errors across models. To validate this observation, we compared the information gap— defined as the information entropy difference between the asymptotic and finite sample size values in Fig. 2a—with the testing errors reported for MACE models trained on these per-molecule dataset splits[22]. The correlation between the two quantities is shown in Fig. 2b, and the information gap curves are shown in Fig. S11. The information gap is a strong predictor of the error in forces, with a Pearson correlation coefficient of 0.89. Even for a constant number of samples (Fig. S12), the information gap explains major variations in force errors for the models, with the ethanol molecule being the only exception to the trend (see Supplementary Text, Sec. A.9 and Figs. S13, S14). This suggests that, in a typical MLIP model, the information gap may relate to a minimum theoretical error that can be achieved across a sampled PES, similar to the lossless compression theorem for information theory.

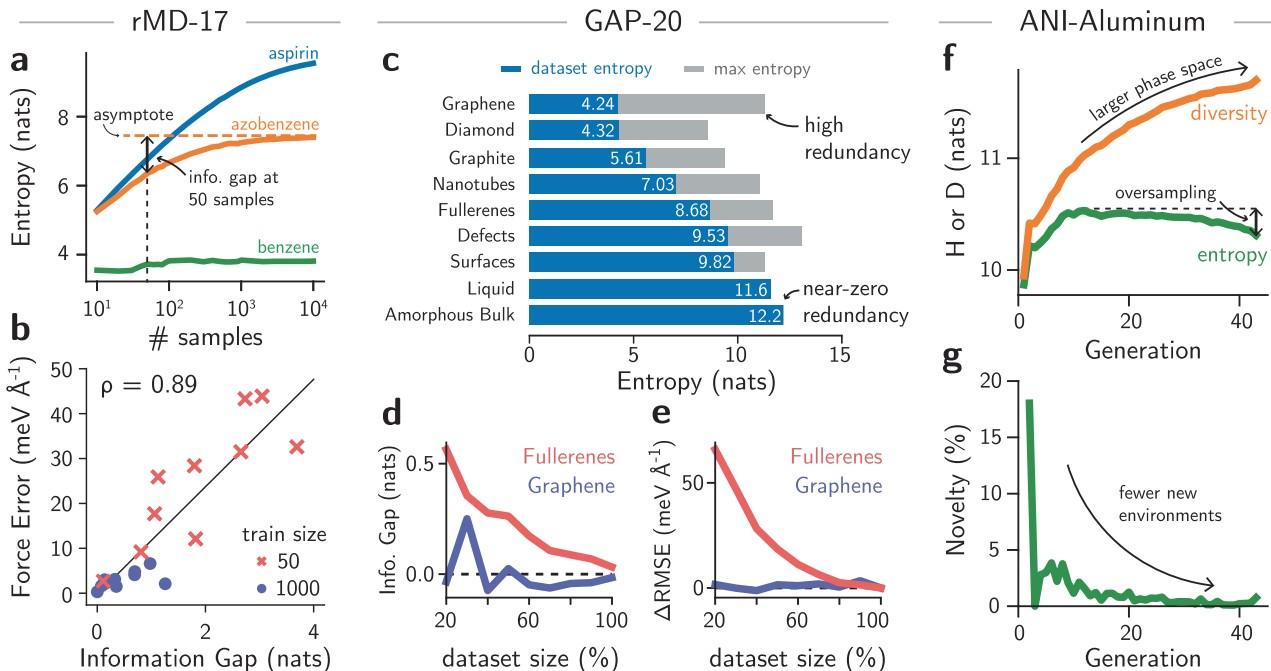

**Fig. 2 | Information entropy measures dataset completeness, compressibility, and sample efficiency in machine learning interatomic potentials.**
**a** Information entropy for three example molecules from the rMD17 dataset as a function of the dataset size. Simpler molecules exhibit lower entropy and converge faster, while more diverse molecules require more samples to converge.
**b** Correlation between the error in predicted forces and the information gap for all molecules in the rMD17[16,38]. The errors were obtained from the original reference for MACE[22]. A circle indicates errors when 1000 samples are used to train the models, and crosses are errors when only 50 samples are used to train the models. $\rho$ is the Pearson's correlation coefficient. **c** Information entropy (blue bars) of selected subsets of the carbon GAP-20 dataset[39]. The maximum entropy is given by $\log n$ (gray bars), where $n$ is the number of atomic environments. The results are sorted by dataset entropy. The numbers are the dataset entropy (in nats).

**d** Information gap obtained by compressing the "Fullerenes" and "Graphene" subsets of GAP-20 by up to 20% of their original sizes. While the information gap of "Graphene" remains close to zero, the one from "Fullerenes" monotonically increases as the dataset size decreases. **e** Test root mean squared error errors relative to the errors obtained when a MACE model is trained on the full subset of GAP-20 ($\Delta$RMSE). The results show that the "Graphene" subset can be compressed by up to 20% of its size without loss of performance, whereas this is not the case for the "Fullerenes" subset. **f** Information entropy ($\mathcal{H}$) and diversity ($D$) for the ANI-Al dataset[31] computed for each generation of active learning. Oversampling of certain phases leads to a total reduction of entropy, as demonstrated by (**g**), showing decreasing novelty in the samples. In this approach, novelty is the fraction of environments showing $\delta\mathcal{H} > 0$ when the dataset of all previous generations are taken as reference. Nevertheless, the diversity of the dataset continues to increase.

Conversely, test errors for molecules such as benzene may be equivalent to the training error of the models, as a near-zero information gap suggests that the training set contains complete information about a given configuration space. As the benchmarks in the literature are performed at a constant number of samples, test errors vary due to differences of information content in each subset, and the information metric can be used to create trade-offs between accuracy and training set sizes.

**Dataset completeness in solid-state systems.** Analogously, this notion of completeness can be useful to post-process existing datasets and quantify redundancy due to sampling and data curation. Within information theory, entropy is used to guide the development of lossless compression algorithms and encoding methods, which is closely related to our dataset reduction goal without loss of information. To demonstrate this approach beyond the rMD17 molecular dataset, we computed the entropy of different subsets of the carbon GAP-20 dataset[39], herein referred to as GAP-20. The comparison between the subset entropy and the maximum possible entropy for a dataset with the same number of environments is shown in Fig. 2c (see Fig. S15). This difference between the maximum possible entropy and the subset entropy, shown with gray bars in Fig. 2c, is complementary to the information gap. Instead of quantifying how much information is needed to reach a converged dataset, a large difference between $\log n$ and the dataset entropy often indicates oversampling in a dataset. In the field of MLIPs, test errors are typically used to quantify saturation of a dataset[39]. However, our information theoretical analysis

provides absolute bounds to the entropy and quantifies the completeness of the dataset without training any model. For example, the difference between the actual information contained in the "Graphene" subset of carbon GAP-20 and the absolute limit given by $\log n$ shows that this subset has large redundancy compared to the "Fullerenes" subset, where the difference between the maximum and actual entropy is smaller. The bounds also illustrate how different datasets can exhibit larger diversity. For example, structures labeled under the "Liquid" and "Amorphous Bulk" categories are maximally diverse (Fig. S15), with environments mostly distinct with the bandwidth used to compute the KDE (0.015 Å$^{-1}$, see Methods). This may be a consequence of both the larger accessible phase space by these amorphous and liquid structures and the original farthest point sampling approach used when constructing the dataset[39].

To illustrate the relationship between information entropy and dataset compression, we computed the entropy curves of different subsets of the GAP-20 dataset. Then, we trained a NNIP based on the MACE architecture[22] on randomly sampled fractions of the subsets, computing test errors as a function of training set size and, thus, entropy. Figure 2d exemplifies this relationship for the labels "Graphene" and "Fullerenes" of GAP-20, which exhibit large (Graphene) and small (Fullerenes) levels of redundancy (Fig. 2c). In the former, datasets as small as 20% of the original one still exhibit entropies around 4.25 nats, similar to the full one. Accordingly, their test errors remain constant across all dataset sizes (Fig. 2e), with a value of 0.96 ± 1.37 meV Å$^{-1}$ for force errors relative to model trained on the full training set. Despite fluctuations in total entropy caused by the

random sampling approach—which depend on unit cell sizes and ordering of structures in the dataset, and become more sensitive at the low-data regime—these results show that our model-free analysis of dataset entropy correctly informed the redundancy of the dataset. On the other hand, the dataset labeled as "Fullerenes" is less redundant, and subset entropies monotonically decrease as the training set size goes down. As expected, the test errors also increase with smaller training set sizes, reproducing known patterns in learning curves of MLIPs (Fig. 2e). Although this example considers only a random sample of data points when "compressing" a dataset, different algorithms can be used in future work to evaluate optimal subsets with maximum entropy for compression of training sets for MLIPs[40]. The method can also be used to evaluate extrapolation conditions or dataset completeness when fast data generation approaches are targeted[41].

**Information efficiency and diversity in active learning loops.** Finally, to exemplify how information theory can be useful to evaluate active learning (AL) strategies in MLIP-driven atomistic simulations, we analyze dataset metrics of the ANI-Al dataset[31], which is a dataset for aluminum constructed by starting from random structures and performing over 40 generations of sampling and retraining with NNIP-driven MD simulations. Figure 2f shows how the entropy varies as new configurations are sampled by the AL. In the initial stages of the active learning, the entropy of the dataset quickly increases, then peaks around generation 12, before subsequently decreasing. To explain this effect, we observe that the increase in diversity of this dataset comes at the cost of oversampling certain regions of the configuration space. In fact, fewer than 5% of the environments sampled after the third round of AL are novel according to our information-theoretical criterion (Fig. 2g). This agrees with the known fact that although MD simulations provide a physically meaningful way to sample new configurations, most sampled configurations are correlated and may be already contained in the original training sets. This may be especially true for large periodic cells where a handful of unknown environments (i.e., $\delta\mathcal{H}>0$) may not be easily separated from the numerous known (or similar-to-known) environments ($\delta\mathcal{H} \leq 0$) that may surround them. To verify that the total coverage of the configuration space still increases, we propose an additional metric of dataset diversity $D$,

$$D(\{\mathbf{X}\}) = \log\left[\sum_{i=1}^{n} e^{\delta\mathcal{H}(\mathbf{X}_i)}\right], \quad (8)$$

that reweights each data point's contribution to the information entropy based on how well-sampled its region of the configuration space is (Supplementary Text, Section A.7). The diversity $D$, in this case, estimates the size of the distribution's support, whereas $\mathcal{H}$ relates to the probability distribution itself. Indeed, Figure 2f shows how the dataset diversity continues to grow even when the entropy decreases. This approach of measuring dataset diversity is related to the concept of "efficiency" in information theory[1] and may be used to propose new ways to sample atomistic configurations or automatically create datasets for MLIPs in the future.

**Model-free uncertainty quantification for machine learning potentials**

When information theory is used to analyze a single dataset, as in the previous section, environments are compared against other environments in the same dataset. However, it is convenient to test the case when reference datasets $\{\mathbf{X}\}$ do not contain a tested sample $\mathbf{Y}$, often leading to $\delta\mathcal{H}(\mathbf{Y}|\{\mathbf{X}\})>0$. In this scenario, we propose that differential entropies can be used as a model-free uncertainty estimator for a given dataset. Whereas uncertainty quantification (UQ) methods for MLIPs usually rely on models[42,43]—i.e., prediction uncertainties are associated to variances in model predictions—we propose instead that an estimate for UQ in MLIPs can be obtained from the data alone. This

approach is similar to Gaussian process regression methods[13,44,45], which compute an uncertainty by inverting a covariance matrix computed for training points, or parametric models on a latent space[46]. Differently from other approaches, however, our method performs a fast non-parametric estimate directly on the atomistic data space. While this approach can be expensive for large datasets, it is easily parallelizable and provides a conservative and deterministic measure that correlates with uncertainties, which are often challenging or expensive to capture within model-based UQ. Our method also makes no assumptions on the generalization power of the model that was trained on these data points, and can be seen as a lower bound of performance (i.e., when generalization is not expected), similarly to what is already performed with Gaussian process regression methods[13,45]. Detaching the model uncertainty from the data uncertainty also allows our framework to be incorporated into a variety of workflows without modifying architectures, loss functions, or increasing the associated cost of training and evaluating the models like NN ensembles[43].

To exemplify how information theory can be used for error prediction in MLIP datasets, we first computed the values of $\delta\mathcal{H}$ for different subsets of the GAP-20 dataset discussed in the previous section. Then, we computed the overlap between one subset $\{\mathbf{Y}\}$ given another reference set $\{\mathbf{X}\}$ of configurations by quantifying the fraction of points $\mathbf{Y}_i \in \{\mathbf{Y}\}$ with $\delta\mathcal{H}(\mathbf{Y}_i|\{\mathbf{X}\}) \leq 0$. Figure 3a exemplifies these overlaps for the "Fullerenes", "Graphite", "Nanotubes", "Graphene", and "Defects" subsets of the dataset (see Fig. S16 for complete results). The results show that environments in the "Graphene" split are mostly contained in the other subsets, with a minimum overlap of 46% between "Graphene" and "Graphite". Though counter-intuitive from the physics perspective, the "Graphite" dataset contains bulk structures which, from the atomic environment perspective, are farther from "Graphene" compared to, say, "Nanotubes" or "Fullerenes." Interestingly, "Nanotubes" contains almost all environments of the "Graphene" dataset, but "Graphene" contains only 53% of the environments in "Nanotubes." Similarly, "Fullerenes" contains a sizeable portion of "Nanotubes", with an overlap of 68%, but not the other way around. This analysis also allows us to identify how each subset is constructed without having to label the structures beforehand. For example, Fig. S16 shows that the "Graphene" subset is also contained by the "Defects" and "Surfaces" datasets. The subsets labeled as amorphous or liquid do not overlap with any of the others, even though their phase space could have been similar depending on their construction method. Finally, large subsets such as "Defects" and "SACADA" contain several parts of the other subsets, largely due to the way they were created. While the discussion here is based on (human) labels attached to subsets of the GAP-20 dataset, this overlap analysis can be used to compare arbitrary pairs of datasets in general, regardless of available labeling. For example, the similarity of test/validation splits can be analyzed to verify whether small errors are due to good model performance in generalization tasks (small overlaps) or simply because the splits are overly similar to the original training set (high overlaps) without any prior information on the structures.

To verify whether overlap between training and testing sets is useful as a predictor of uncertainty and error metrics in actual models despite not making assumptions about their performance, we trained models based on the MACE architecture to each one of the subsets of GAP-20 in Fig. 3a, then tested the models on the other splits. Figure 3b shows the test errors obtained from such training-testing splits. When models are tested on subsets exhibiting large overlap with their training set, all of them perform well, with normalized errors below 10%. On the other hand, errors tend to be much larger when the overlap between train and test sets are small, sometimes surpassing 100% error (see also Fig. S17). These results show a power law for distinct train/test sets with clear anti-correlation between the error and overlap.

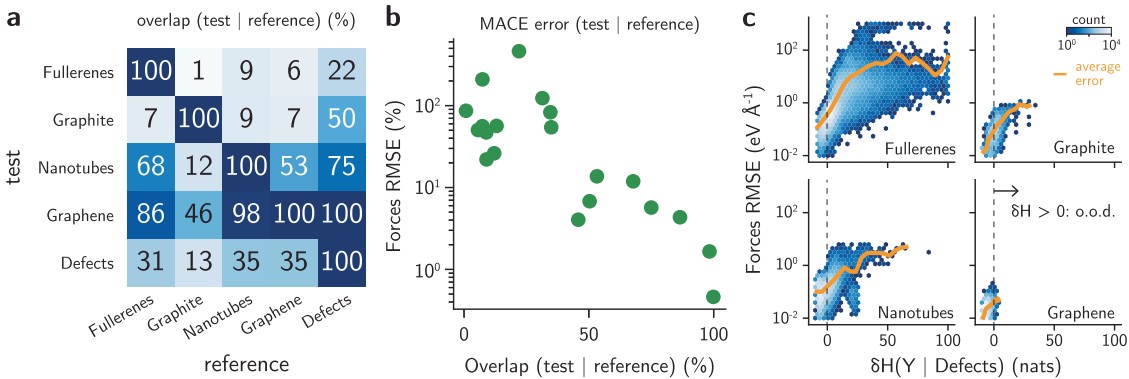

**Fig. 3 | Information entropy quantifies overlaps between datasets and predicts errors in machine learning potentials. a** Overlap between test and reference sets for the GAP-20 carbon dataset[39]. Only a subset of the data is shown for clarity (see Fig. S16 for complete matrix). **b** Test errors (in %) of a MACE model[22] trained on one of the subsets of the GAP-20 dataset shown in (**a**), and tested on the other subsets. Each point corresponds to a single (train, test) pair. Models with higher overlaps between train and test sets exhibit substantially smaller errors. **c** Correlation between force root mean squared errors (RMSE, in eV Å$^{-1}$) and the differential entropy $\delta\mathcal{H}(\mathbf{Y}|\text{Defects})$ for environments $\mathbf{Y}$ from different datasets computed using the dataset "Defects" as reference. The RMSE corresponds to the error of models trained on the "Defects" subset of GAP-20 and tested on the other environments. The average RMSE (orange line) increases with higher $\delta\mathcal{H}$. For "Fullerenes", The values of $\delta\mathcal{H}$ were truncated to 100 nats for clarity.

To further this observation beyond dataset averages and towards an environment-based uncertainty metric, we evaluated the MACE model trained on the "Defects" split of the GAP-20 dataset against the other other four splits, exhibiting increasing overlaps with "Defects": "Fullerenes" (22% overlap), "Graphite" (50%), "Nanotubes" (75%), and "Graphene" (100%) (Fig. 3a). Force errors were then evaluated for this model and correlated with the values of $\delta\mathcal{H}$, for each environment, as shown in the distributions of Fig. 3c. For environments where $\delta\mathcal{H}>0$, the RMSE is often above 0.1 eV Å$^{-1}$. On the other hand, when $\delta\mathcal{H}\leq 0$, errors are typically lower than that. To demonstrate that higher values of $\delta\mathcal{H}$ usually lead to higher errors beyond the correlation plots, we computed the running average of RMSE for a constant window size of $\delta\mathcal{H}$. Figure 3c shows that average errors continue to increase as the values of $\delta\mathcal{H}$ also increases, showing that points further away from the training set tend to exhibit higher errors. On the other hand, points slightly outside of the known domain, thus with positive but near zero $\delta\mathcal{H}$, often show average errors comparable to the ones in the training set. Interestingly, Fig. 3c also shows that force errors continue to decrease as $\delta\mathcal{H}$ becomes more negative. This correlates with the idea that unbalanced datasets bias the training process and end up minimizing the loss for data points with higher weight (i.e., with more negative $\delta\mathcal{H}$). The same observation is valid for the maximum error within each range of $\delta\mathcal{H}$ (Fig. S18), illustrating how negative entropy values lead to small errors provided that errors are small everywhere in the training set. Furthermore, because the uncertainty threshold $\delta\mathcal{H}>0$ is guaranteed by the theory to denote extrapolation (Supplementary Text, Section A.4), our uncertainty measure detects points outside of the training domain without the need for additional calibration or empirically fitted parameters. Whereas this measure does not have the same units of error, similar to other density estimation strategies in MLIPs[43,46], an actual quantification of error values from the computed $\delta\mathcal{H}$ can be performed with calibration or conformal prediction methods[42,43]. In this case, error estimates can be obtained directly from the computed values of $\delta\mathcal{H}$ (see Fig. S19 and Supplementary Methods) and are demonstrated to correlate well with the actual errors from the model. Thus, our information theoretical approach provides a robust, model-free alternative to estimating uncertainties in MLIPs.

## Information theory explains chemical and error trends across the TM23 dataset

To demonstrate that the methods above can be combined to analyze other datasets, we used our information entropy formalism to explain trends in the TM23 dataset[47], which contains structures and properties of elemental transition metals across temperatures. Owen et al. demonstrated that trends in force errors of models fitted to separate elements in this dataset are challenging to explain from elemental properties alone and are better described by many-body interactions due to the electronic structure of the metals[47]. We show that these trends in electronic structure and model errors agree with the theoretical analysis in this work by quantifying the entropy, diversity, information gap, and dataset overlaps for each transition metal in the TM23 dataset. Figure 4a depicts the information entropy for each elemental subset of TM23 using the "full" data split, i.e., including information about all three temperatures of the dataset. The entropy table, arranged like the d-block of the periodic table, indicates with brighter colors the elemental subsets of TM23 with higher information entropy. The values immediately resemble the original relative force errors for NequIP models trained to the full dataset[47], shown in Fig. 4b, where coinage metals exhibit smaller entropy and smaller errors, and early transition metals display higher information entropy and higher force errors (see also Figs. S20–S22). At the same time, Zn, Cd, and Hg are outliers not only in force errors, as pointed out by Owen et al.[47], but also contain higher information entropy. As entropy and diversity are related, Fig. S21a shows that more diverse datasets often lead to higher relative errors in force prediction, with these quantities having a Pearson correlation coefficient of $\rho = 0.74$. These results are also correlated with the d-band center of the metals (excluding group 12), showing that differences in the electronic structure are related to the information contents in the dataset (Fig. S21b). As the electronic structure and the accessible phase spaces are strongly connected, our information entropy is measuring the indirect consequence of diversity in configurations that, in essence, are governed by the electronic structure of the materials. Qualitative visualization of Fig. 4a,b may also explain why group VI elements exhibit a lower error compared to group V or VII ones, with the information entropies recovering smaller information contents in Mo compared to Nb or Tc, and similarly for W compared to Ta or Re. To verify if dataset incompleteness (i.e., low number of samples) also impacted the force errors in the TM23 dataset, we computed the information entropy as a function of the number of environments in the training set, as exemplified in Fig. S23, and found the information gap to be strongly correlated with the diversity and entropy of the datasets (Fig. S22). This finding recovers the expected trends of information-rich datasets being "harder to learn" (e.g., Os or Ti), while datasets with less information such as Cu or Ag exhibit smaller errors. Thus, more diverse phase spaces may require more samples to ensure lower information gaps and (possibly) lower errors.

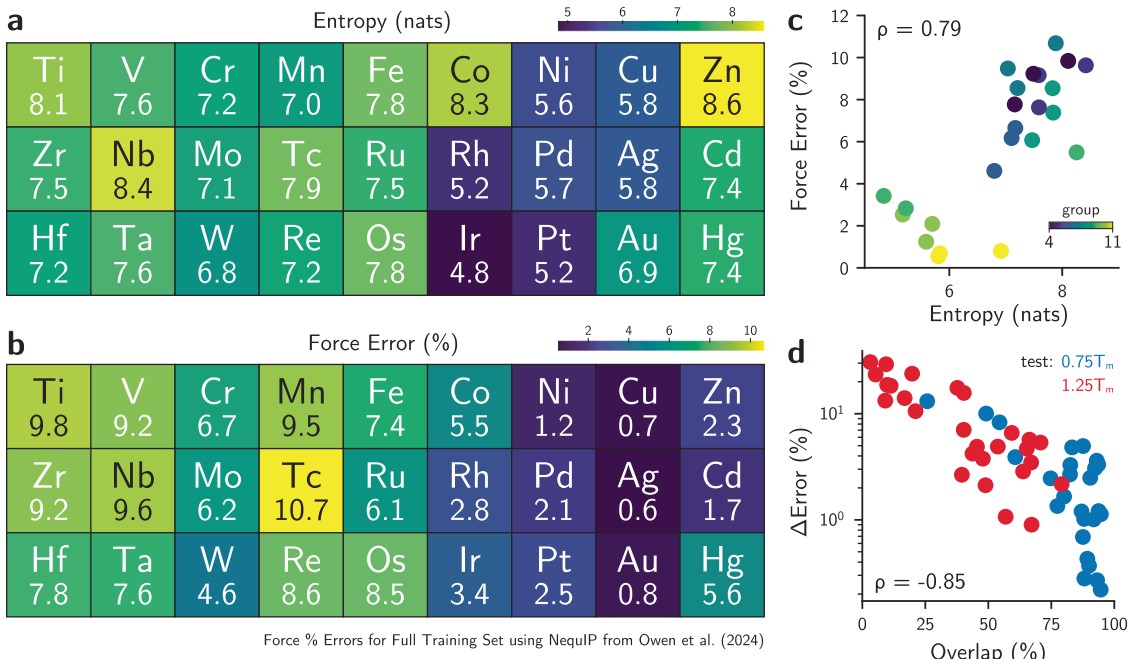

Force % Errors for Full Training Set using NequIP from Owen et al. (2024)

**Fig. 4 | Information theoretical quantities correlate with error and chemical trends in the TM23 dataset. a** Information entropy of the full TM23 training set for each element[47]. **b** Force errors (in %) for NequIP models[21] trained on the full training set, obtained from Owen et al.[47]. **c** These two quantities exhibit strong correlation, as indicated by the Pearson correlation coefficient of $\rho = 0.79$ for transition metals with incomplete d-shell. **d** The difference between the final forces error (in %) and the initial forces errors (in %) (denoted as $\Delta$Error) is explained by the dataset overlap obtained by computing the differential entropies $\delta\mathcal{H}(0.75T_m|0.25T_m)$ or $\delta\mathcal{H}(1.25T_m|0.25T_m)$, as demonstrated by the Pearson correlation coefficient of $\rho = -0.85$. Red and blue dots indicate error differences for models trained on the "cold" subset of TM23 (sampled at 0.25 $T_m$, where $T_m$ is the melting temperature) and tested on the "warm" (0.75 $T_m$) and "melt" (1.25 $T_m$), respectively.

To study model generalizability, the authors of the TM23 dataset also proposed a test of model transferability across temperatures using the sampled structures. Specifically, models trained on low-temperature samples (subset "cold") are tested against high-temperature samples (subsets "warm" and "melt"), and the resulting errors are compared. Interestingly, the original benchmark demonstrated that different elemental datasets exhibit varying levels of transferability, with good extrapolation behavior in platinum-group metals and coinage metals, but poor transferability in early transition metals such as Re or Cr. Using the information-based UQ method described in the previous section, we explain these trends in model performance solely from the analysis of dataset overlap using the differential entropy $\delta\mathcal{H}$. Using the low-temperature samples as reference dataset, we computed the values of $\delta\mathcal{H}$ (warm | cold) or $\delta\mathcal{H}$ (melt | cold) for each element, and computed the overlap by measuring the fraction of the dataset with $\delta\mathcal{H} \leq 0$. The comparison between error differences—i.e., the degradation of the model performance relative to the model trained on the full elemental dataset—and the dataset overlaps is shown in Fig. 4d. The strong anti-correlation between the error differences and the dataset overlaps, with a correlation coefficient $\rho = -0.85$, further confirms the results discussed above for the model-free UQ analysis. Interestingly, it also explains why transferability trends shift as we move from the "warm" (0.75 $T_m$) to the "melt" (1.25 $T_m$) datasets. For instance, Re has the worst error in the transferability test from "cold" to "warm", as also shown in Fig. S24, but not in the "cold" to "melt" test, where models trained to the Zr, Hf, and Cr datasets are the worst-performing ones. Our analysis shows that the overlaps between the "melt" and "cold" subsets for the latter three examples indeed drop to levels smaller or equal to Re (i.e., nearly zero overlap, see Table S1), whereas the overlap between their "warm" and "cold" subsets is substantially larger ( ~ 50%) than the one for Re. Our analysis also revealed meaningful information on outliers that remained unexplained in the original TM23 results. For example, Tc has the worst force error (in %) when models are trained to the full

dataset (Fig. 4b), but better transferability across temperatures. Our analysis explains these results by the higher overlap in phase spaces between the "cold" and "warm" (92%) and "cold" and "melt" (79%) for Tc compared to the other metals, suggesting that the process of heating and melting when constructing the dataset for Tc did not vary the phase space as much compared to other elements. Though much more detailed analysis can be performed across the chemical trends, the results shown here demonstrate that performance in MLIPs can be well-approximated even in the absence of models, further strengthening the usefulness of the information-based analysis for several tasks in atomistic ML.

**Information-based detection of outliers and rare events in atomistic simulations**

Our information theoretical method can be further employed to detect outliers (on-the-fly) within a large-scale production MLIP-driven MD simulation and is not restricted to only comparing static datasets. To demonstrate this concept, we produced an MD trajectory of dynamically strained Ta using a supercell containing approximately 32.5 million atoms and a SNAP potential[15] fitted to an EAM potential (see Methods). This choice of potential was made so we could have access to the ground-truth energies and forces even at very large scales, thus allowing us to benchmark our outlier detection approach. In such large models, obtaining uncertainty estimates of energy/force predictions can be challenging even at the postprocessing stage, especially if it requires re-evaluating predictions with several models, such as with an ensemble-based UQ approach. Furthermore, uncertainty thresholds may not be well-defined for models such as SNAP, where the choice of weights, training sets, and hyperparameters can lead to substantial variations of model performance[33]. Finally, ML-driven simulations of periodic systems may fail in completely different ways compared to simpler molecular systems[24,25], where bond lengths and angles are sometimes sufficient to detect extrapolation behavior.

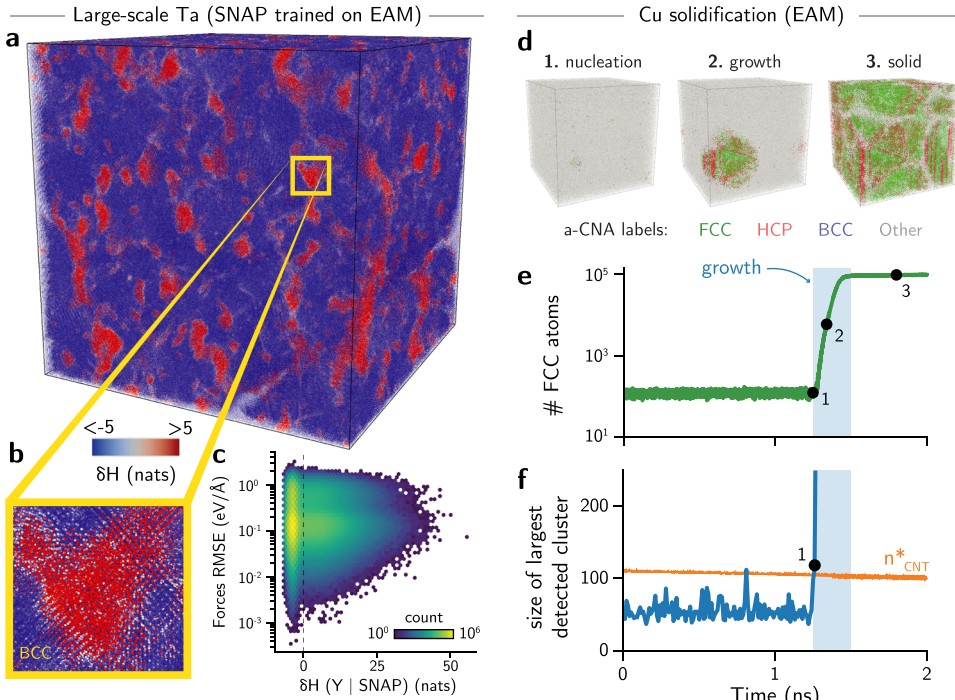

**Fig. 5 | Differential entropies detect outliers and rare-events in large atomistic simulations. a** Visualization of a 32.5M atom snapshot of body-centered cubic (BCC) Ta simulated with the Spectral Neighbor Analysis Potential (SNAP)[15]. Colors represent the values of the estimated differential entropy $\delta\mathcal{H}$, with blue atoms indicating environments reasonably within the training set ($\delta\mathcal{H}<0$) and red atoms indicating environments outside of the training set ($\delta\mathcal{H}>0$). Values of $\delta\mathcal{H}$ were truncated to the range $[-5, 5]$ to facilitate the visualization of divergent colors. The dataset used to train the SNAP models was constructed using an embedded atom method (EAM) classical potential for tantalum. **b** Example of high-uncertainty region encountered during the simulation. The formation of a disordered, non-BCC phase (red) in the simulation leads to unphysical behavior in the trajectory. **c** The unphysical behavior cannot be identified only by the root mean squared errors (RMSE) in forces. Even outside of its known domain, the SNAP model exhibits errors within the range of systems within the training set. However, the differential entropy $\delta\mathcal{H}(\mathbf{Y}|\text{SNAP})$ of probe environments $\mathbf{Y}$ (computed using the SNAP training set as a reference) can be used to distinguish between in-distribution and out-of-

distribution environments. The number of environments in each region is shown by the color scale, with brighter colors indicating exponentially denser regions of the error-$\delta\mathcal{H}$ space. **d** Visualization of a solidification trajectory for copper, simulated using an EAM potential, during its nucleation, growth, and solid stages. Face-centered cubic (FCC), hexagonal close-packed (HCP), and BCC phases are shown with green, red, and blue colors, respectively. Non-identified phases are represented in gray. **e** Number of FCC atoms derived from the molecular dynamics (MD) simulation with the adaptive common-neighbor analysis (a-CNA) method[62]. The shaded blue area indicates the time window where crystal growth is observed. The critical nucleus is observed around 917 K. The black dots indicate the frames corresponding to nucleation, growth, and final solidified system visualized in **a**. **f** Largest cluster size in the simulation box, obtained by grouping atoms with $\delta\mathcal{H} \leq 0$. The orange line represents the estimated critical nuclei sizes $n^*_{\text{CNT}}$ estimated using the values of interfacial energy, melting temperature, and melting enthalpy for the potential, and the average undercooling for each time step, according to the classical nucleation theory (CNT).

Using the values of $\delta\mathcal{H}$ computed for each environment (i.e., atom center) in the 32.5M atom system against the original training set, we analyzed a snapshot of the Ta MD trajectory to identify possible anomalies due to extrapolation during the simulation. Figure 5a shows that about 13% of the environments at this time step exhibit $\delta\mathcal{H}>0$ (red atoms), some of which are as large as $\delta\mathcal{H}=55.8$ nats (see Fig. S31), showing that a substantial number of environments are outside of the training domain of the potential. This particular simulation was initialized with a monocrystalline BCC crystal of Ta seeded with dislocations, following the approach from Zepeda-Ruiz et al.[48]. After propagating the trajectory with MD simulation, much of the cell retained its body-centered cubic (BCC) structure, which is well represented in the training set (and thus appears as the blue colored atoms), but the simulation also proceeded to form amorphous-like phases (Fig. 5a, b) that are unexpected in such trajectories and are detected as the out-of-domain samples (red). Although the model prevents obvious nonphysical configurations, such as overlapping atoms, distinguishing between such model failures and new physical phenomena (such as finding reasons for strain hardening) in these simulations is challenging without outlier detection tools. To illustrate the difficulty of identifying a possible model failure, we computed the ground truth forces for the atomistic system from the interatomic potential that originated the labels for the SNAP training set, and analyzed the errors

of the predictions as a function of $\delta\mathcal{H}$ for each configuration (Fig. 5c). Surprisingly, the results show that the SNAP model under investigation does not exhibit high extrapolation errors in forces. Rather, the RMSE of forces are within the same range of errors for environments having $\delta\mathcal{H}>0$ and $\delta\mathcal{H}<0$. Importantly, this is not an artifact of our uncertainty measure. Instead, the formation of the amorphous phase is likely due to lower predicted energies compared to the true energies for these out-of-domain configurations. Therefore, even access to the ground truth forces does not allow the classification of a trajectory as "failed" within these constraints. Instead, it would rely on human inspection of the trajectory, which can be subjective and is prone to observer error. These issues are already challenging to anticipate without outlier detection methods, and even more in simulations where the ground truth is not accessible (e.g., training sets constructed with DFT). On the other hand, our differential entropy detects these outliers without the need for a calibrated threshold, providing a model-free estimate for extrapolation and an "early warning" that can be used for model augmentation, re-evaluation, or retraining.

Beyond ML robustness and UQ, finding outliers in atomistic simulations is also of importance for simulations involving phase transitions, emergent complex phenomena, or other out-of-equilibrium events during MD trajectories. Because our information-theoretical method is model-independent, we illustrate in particular

how the approach can be used to detect rare events in MD trajectories. As a model system, we analyzed the nucleation of copper simulated using the potential from Mishin et al.[49] in a supercell containing nearly 300,000 atoms with undercooling of approximately 420 K at 1 bar. The main stages observed during the trajectory included: (1) nucleation of a crystal from the melt; (2) crystal growth regime; and (3) solidified system with residual liquid and grain boundaries (Fig. 5d). The nucleation event was obtained by gradually decreasing the temperature of the molten system over 2 ns (Fig. S26). Classification of the atomic environments using the traditional common neighbor analysis (CNA)[50] reveals the appearance of a dominant FCC phase in the second half of the simulation, with rapid growth for about 100 ps, and a plateau in later stages (Fig. 5e).

This process of nucleation and growth is typically explained by approaches such as the classical nucleation theory (CNT), which uses near-equilibrium assumptions to model an out-of-equilibrium event. The CNT proposes that there is a statistical distribution of solid clusters in the liquid phase, and the nucleation event happens when an outlier cluster of critical size is formed in a probabilistic event. However, observing or detecting these outlying critical nuclei directly from atomistic simulations can be challenging. Discrete classification of the solid phases, e.g., using variations of the CNA, typically prevents identification of the structure of the liquid phase, as very few clusters are found and classified correctly as solid-like when they are found in the melt (see Fig. S27). To verify if our information theoretical model can detect outliers as in nucleation theory, we computed the values of $\delta\mathcal{H}$ for each frame in the simulation using an MD trajectory of pure FCC copper as reference. To ensure a conservative estimate of what defines a "solid-like" cluster, the reference dataset was taken from a MD simulation at a low temperature of 400 K and pressure of 1 bar in the NPT ensemble. Then, frames of the original solidification trajectory were compared against snapshots of this low-$T$ reference MD. Instead of considering environment labels assigned by an algorithm, we detected nuclei in the melt by finding environments with high overlap with the phase space of the pure solid. In practice, this means that subcritical nuclei can be identified by taking environments $\mathbf{X}_{melt}$ such that $\delta\mathcal{H}(\mathbf{X}_{melt}|\{\mathbf{X}_{solid}\})<0$. Using this method, we obtained the number and size of clusters by using a graph theoretical analysis, where nodes are environments with $\delta\mathcal{H} \leq 0$, edges connect nodes at most 3 Å apart, and clusters are the sets of connected components (Methods). Then, we analyzed the largest cluster size among all extracted subgraphs (Fig. 5f). With this analysis, we observed the largest cluster size is typically below 100 atoms until the nucleation event, when it reaches the value of 114 atoms. In contrast, the CNA method recovers a maximum of 170 (mostly isolated) FCC environments within the entire simulation box and across all pre-nucleation frames (Fig. S27). To compare this with predictions from the CNT, we calculated the critical nucleus size given the average, time-dependent undercooling in the simulation. The melting enthalpy and temperature obtained from the simulation's EAM potential were used to perform such estimate, as calculated by Mendelev et al.[51] The solid-liquid interfacial energy was also computed for the potential by Zepeda-Ruiz et al.[52] and determined to be equal to 0.177 J/m². This value is in excellent agreement with experiments, which range between 0.177 and 0.221 J/m² for Cu[53–55]. The critical nucleus size estimated at each timestep (given the undercooling) and the simulated values for the Cu potential is shown in Fig. 5f in orange color. The results demonstrate that nucleation happens when the largest cluster identified by our information theoretical method crosses the threshold of minimum cluster size for nucleation. Prior to the nucleation event, only a single other frame intersects the threshold of minimum cluster size for nucleation. Visualization of the cluster indicates that the graph at that frame is better approximated as two nuclei rather than a single, compact critical nucleus (Fig. S25, see also Sec. A.10 in the Supplementary Text), which may be an artifact of the graph-theoretical approach used to identify the clusters.

Nevertheless, this demonstrates that our information-theoretical approach can be used to detect rare events in atomistic simulations, including cases where ground-truth values (e.g., force errors) or existing classification methods (e.g., a-CNA) lack sensitivity to predict.

## Discussion

Our results show how an information theoretical approach can be used in a range of problems in atomistic modeling. By computing distributions of atom-centered representations from simulations, our information-based analysis of datasets and uncertainty explains multiple results within learned interatomic potentials and atomistic simulations in general, with a fast model-free approach. In particular, we showed how information and diversity content in a dataset can be predicted directly from a training set, explaining error trends in MLIPs, rationalizing dataset compression, estimating extrapolation errors, and detecting outliers or rare events in MD trajectories. Moreover, because information entropy provides a quantitative estimation of "surprise" of a random variable, we proposed its use as a UQ metric for ML-driven atomistic simulations and showed it can be used for outlier detection even for large atomistic systems. As this strategy does not depend on models, it can be adapted to any MLIP (or any fitted potential) to provide useful uncertainty metrics.

Interestingly, beyond its use in MLIPs and atomistic datasets, we observed that the information entropy of atom-centered descriptor distributions qualitatively recovers the configuration component of thermodynamic entropy differences (i.e., entropy differences due to vibration, but not momenta or composition) in several cases. For example, we obtained surprisingly accurate predictions of entropy differences for several elemental systems by adjusting the bandwidth parameter according to a physics-based rule, allowing us to compute phase diagrams at high pressures and temperatures (see Supplementary Text A.11, Figs. S29 and S30). Whereas the parallels between information and thermodynamic entropies are well-known, there is no guarantee that descriptor distributions should represent the actual thermodynamic entropies beyond simple configurational components. Nevertheless, the coincidence between information and the configurational component of thermodynamic entropies will be investigated in more detail in upcoming works.

In the future, several strategies can further generalize this method. For example, the approach does not explicitly account for element types due to the choice of representation. For simple molecular systems, bonding patterns (e.g., valence rules) sometimes map distributions of atomic environments to different parts of the information entropy space due to the construction of the $k$-nearest neighbors descriptor based on interatomic distances. However, for inorganic crystals, this approximation is not valid, and may have to be incorporated into the approach to account for true configuration entropies, as in alloys. Furthermore, as the feature space of several atom-centered representations may not be injective[56], the information entropy depends on the choice of descriptor. Nevertheless, the formalism presented here is general and can be adapted to other representations by changing the metric space and kernel function to other appropriate choices. The explicit dependence of the information entropy for atomic environments according to the kernel function and descriptors will be investigated in future work. Finally, whereas the current computational implementation is sufficient for the analysis of tens of millions of environments, improvements in parallelization and hardware utilization can allow the approach to scale towards a real-time UQ for MD. Our analysis of the parallelized and approximated results for the large-scale Ta system shows that the information entropy can be computed for large systems even with few resources (see Supplementary Text, Sec. A.12). While computation of kernel matrices already have been implemented in the code, faster computation of descriptors, multi-node parallelization, or better approximate nearest neighbors computation can be implemented in future versions

of the code, allowing greater scaling in computing kernel density estimates and their resulting information contents.

In summary, in this work, we proposed a strategy for quantifying information in atomistic simulations and ML based on information theory. By performing a kernel density estimation over distributions of atom-centered features, we obtained values of information entropy that: (1) rationalize trends in testing errors for machine learning potentials across multiple datasets, relating model performance to information contents; (2) quantify compressibility and sampling efficiency for atomistic datasets based on information theory; (3) provide a model-free uncertainty quantification approach for atomistic ML; and (4) allow for outlier detection in large production simulations from MLIPs or general atomistic datasets. These contributions are demonstrated with numerous examples, from known benchmarks from the MLIP literature, a solidification trajectory, and a simulation of a system containing about 32.5M atoms. As increasingly accurate and scalable ML models are proposed for atomistic simulations, this work provides a toolkit that can be used to improve these efforts in several thrusts, including evaluating information contents in datasets, assessing the convergence of active learning loops, guiding the curation of more diverse datasets, and detect anomalies in ML-driven simulations. Additional developments in atomistic information theory can continue to translate developments in machine learning and other statistical methods into faster and more accurate materials modeling.

## Methods
### Information entropy and QUESTS method
**Representation:** the representation of atomic environments was computed as described in Section "Formulation of an atomistic information entropy" of the main text and Section A.1 of the Supplementary Text. Throughout this work, a number of $k = 32$ neighbors was used to represent the atomic environment, with a cutoff of 5 Å. This range of neighbors and cutoffs approximates the hyperparameters typically used in MLIPs, where cutoffs of 5–7 Å are employed. Whereas changing the values of $k$ and cutoff influences the metric space by making environments increasingly similar (at lower $k$ or radii) or dissimilar (at higher $k$ or radii), we observed that the trends in entropy remained consistent throughout the examples in this manuscript (see also mathematical derivation in the Supplementary Text, Sec. A.1). To accelerate the calculation of the representation, the code that computes the descriptors was optimized using Numba[57] (v 0.57.1) and its just-in-time compiler. For periodic systems, the feature vectors were created by adapting the stencil method for computing neighbor lists and parallelizing the creation of features across bins.

**Information entropy:** the information entropy of descriptor distributions was computed as described in Section "Formulation of an atomistic information entropy" of the main text and Section A.2 of the Supplementary Text. Throughout this work, the natural logarithm was used for the entropy computation, which scales the information to natural units (nats). The bandwidth was selected by computing the distance between two FCC structures with lattice parameters 3.58 and 3.54 Å (1% compressive strain), leading to a bandwidth of 0.015 Å$^{-1}$ (see also Sec. A.6 of the Supplementary Text). For the computation of the differential entropy $\delta\mathcal{H}$, the bandwidth and units adopted are the same as the information entropy.

**Entropy asymptotes:** the asymptotic behavior of entropies in the learning curves of Figs. 2a and S10 was obtained by fitting a function of the form

$$f(N) = a - b \exp\left[-c(\log N)^2\right], \tag{9}$$

with $a$, $b$, $c$ non-negative parameters obtained from the entropy curve as a function of training set size $N$. The first and last three points were discarded during the fitting process. The fit was performed using a non-linear least squares method implemented in SciPy[58] (v. 1.11.1). This

functional form was found to closely approximate the curves shown in Figs. 2a, 4e, S10, and S23.

**Entropy of the TM23 dataset:** The entropy was computed for a rescaled dataset to avoid issues with samples at different temperatures and densities, thus facilitating comparisons across metals of different densities using a constant bandwidth of 0.015 Å$^{-1}$. The original distributions of atomic volumes (i.e., the reciprocal of densities) for the TM23 dataset is shown in Fig. S20a. The final rescaled volume was adopted as the median volume of all elemental datasets, and equal to 15.8 Å$^3$/atom.

### Molecular dynamics simulations
All MD simulations were performed using the Large-scale Atomic/ Molecular Massively Parallel Simulator (LAMMPS) software[59] (v. 2/ Aug./2023). All simulations were performed using a 1 fs time step, except when stated otherwise.

**Cu solidification:** the solidification trajectory of copper was simulated using the EAM potential from Mishin et al.[49] A $42 \times 42 \times 42$ supercell of FCC copper (296,352 atoms) was simulated above the melting point to produce the structure of the liquid, then cooled to 924 K. Starting at the temperature of 924 K, the system was cooled to 914 K over the course of a 2 ns-long simulation in the NPT ensemble with the Nosé-Hoover thermostat and barostat[60,61] implemented in LAMMPS. Damping parameters for the temperature and pressure were set to 0.1 and 3.0 ps, respectively, a 2 fs time step was used for the integrator, and constant pressure of 1 bar. Over the trajectory, the number of FCC atoms was computed using the common neighbor analysis (CNA) implemented in LAMMPS[50]. The adaptive CNA variant[62] was used to identify clusters in Fig. S27.

**Large-scale Ta simulation:** The atomistic configuration with "amorphous-like" substructures (Fig. 5) used in benchmarking performance of our information-based detection of structural anomalies resulted from a large-scale MD simulation of crystal plasticity in body-centered-cubic metal Ta. The simulation was performed using a SNAP potential fitted to the dataset of the original SNAP potential[15]. However, rather than using the DFT ground-truth reference values of energies, forces and stress in the original fitting dataset, all the same quantities were re-computed using an inexpensive interatomic potential of the embedded-atom-method (EAM) type for tantalum[63]. Given that both SNAP and EAM simulations can be performed at scales large enough to perform simulations of metal plasticity of the kind described in Zepeda-Ruiz et al.[48], the intention was to observe if a SNAP potential fitted to such a proxy training dataset could reproduce plastic strength predicted by the proxy potential itself. The SNAP simulation considerably diverged from the proxy EAM simulation both in predicted plasticity behavior and in producing the "amorphous-like" regions that never appeared in the proxy EAM simulation. Simulations were performed exactly as reported by Zepeda-Ruiz et al.[48], by using infinite crystal with periodic boundary conditions, with an initial aspect ratio of 1:2:4 ($L_x$: $L_y$: $L_z$), with dimensions equal to $L_x \times L_y \times L_z = 128a_0 \times 256a_0 \times 512a_0$ containing approximately 32.5 million atoms and oriented along the principal [100], [010], [001] axes of the body-centered-cubic lattice, respectively. $a_0 = 0.33$ nm is the lattice constant of tantalum at ambient conditions. The crystal was compressed either at a constant strain rate along the [001] axis ($L_z$), with its two lateral dimensions $L_x$ and $L_y$ allowed to equilibrate the "xx" and "yy" components of the mechanical stress near zero. Temperature was maintained constant at 300 K using a Langevin thermostat with the damping parameter set to 10 ps.

### Machine learning potential
**MACE architecture:** the ML force fields for GAP-20 in this work were trained using the MACE architecture[22]. We used the MACE codebase available at https://github.com/ACEsuit/mace (v. 0.2.0). Two equivariant layers with $L = 3$ and hidden irreps equal to

`64 × 0e + 64 × 1o + 64 × 2e` were used as main blocks of the neural network model. A body-order correlation of $\nu = 2$ was used for the message-passing scheme, and the spherical harmonic expansion was limited to $\ell_{max} = 3$. Atomic energy references were derived using a least-squares regression from the training data. The number of radial basis functions was set to 8, with a cutoff of 5.0 Å.

**MACE training:** the MACE model in this work was trained with the AMSGrad variant of the Adam optimizer[64,65], starting with a learning rate of 0.02. The default optimizer parameters of $\beta_1 = 0.99$, $\beta_2 = 0.999$, and $\varepsilon = 10^{-8}$ were used. The exponential moving average scheme was used with weight 0.99. In the beginning of the training, the energy loss coefficient was set to 1.0 and the force loss coefficient was set to 1000.0. The learning rate was lowered by a factor of 0.8 at loss plateaus (patience = 50 epochs). After epoch 500, the training follows the stochastic weight averaging (SWA) strategy implemented in the MACE code. From there on, the energy loss coefficient was set to 1.0 and the force loss coefficient was set to 100.0. The model was trained for 1000 epochs. A batch size of 10 was used for all models, except in the Defects subset of GAP-20, for which the batch size was adopted as 5. Each dataset was split randomly at ratios 70:10:20 for train/validation/test. The best-performing model was selected as the one with the lowest error on the validation set.

**Error normalization:** in Fig. 3b, the forces RMSE (in eV/Å) was normalized by the average force in the test set to account for different distributions of forces in different splits of GAP-20. This strategy is similar to the original adopted in Fig. 4, whose values of normalized error (in %) were obtained from the original work[47].

## Uncertainty quantification

**Novelty of an environment:** a sample $\mathbf{Y}$ is considered novel with respect to a reference set $\{\mathbf{X}\}$ if $\delta\mathcal{H}(\mathbf{Y}|\{\mathbf{X}\}) > 0$. Therefore, the novelty of a test dataset $\{\mathbf{Y}\}$ with respect to $\{\mathbf{X}\}$ is computed as the fraction of environments $\mathbf{Y}_i \in \{\mathbf{Y}\}$ such that $\delta\mathcal{H}(\mathbf{Y}_i|\{\mathbf{X}\}) > 0$. On the other hand, the overlap between a test dataset $\{\mathbf{Y}\}$ with respect to $\{\mathbf{X}\}$ is the fraction of environments $\mathbf{Y}_i \in \{\mathbf{Y}\}$ such that $\delta\mathcal{H}(\mathbf{Y}_i|\{\mathbf{X}\}) \leq 0$. Larger positive values of $\delta\mathcal{H}$ imply that the test point $\mathbf{Y}_i$ is further away from the training set $\{\mathbf{X}\}$.

**Novelty in active learning:** specifically in Fig. 2g, the novelty of sampled configurations at generation $n > 1$ is obtained by computing the differential entropy $\delta\mathcal{H}$ with respect to the complete dataset at generation $n - 1$.

**Correlations between error and $\delta\mathcal{H}$:** Force errors in Fig. 3d were computed by taking the norm between the predicted and true force for each atom, thus assigning a single error per environment. To average the errors for each $\delta\mathcal{H}$, as shown in Fig. 3e, we binned the values of $\delta\mathcal{H}$ in 20 bins of uniform length $\ell$. Then, for each bin, we averaged the errors for all points within $0.75\ell$ of the center of the bin. This creates a running average effect for the errors, reducing the effect of discontinuities with small displacements of bin centers. At the same time, the bin length $\ell$ is determined by the range of the values of $\delta\mathcal{H}$.

## Classical nucleation theory analysis

**Critical cluster size:** following known results from the classical nucleation theory (CNT), the critical cluster size $r^*$ of a mono-component, spherical cluster in a melt is given by

$$r^* = \frac{2\gamma_{SL}T_m}{\Delta H_m \Delta T},$$

where $\gamma_{SL}$ is the interfacial energy between the solid and liquid, $T_m$ is the melting temperature, $\Delta H_m$ is the latent heat of melting, and $\Delta T$ is the undercooling. For the solidification of copper, experimental values of $\gamma_{SL}$ range between 0.177 and 0.221 J/m$^2$[53–55]. Whereas the experimental melting temperature at 1 bar is 1357.77 K, with latent heat equal to 13.26 kJ/mol, we used the values determined for the potential, with

$T_m = 1323$ K and $\Delta H_m = 11.99$ kJ/mol[51]. The ranges of critical cluster sizes in Fig. 5f were obtained by assuming a spherical cluster and an atomic volume of 12.893 Å$^3$/atom obtained from the simulations.

**Graph-theoretical determination of clusters:** As classification methods such as (a-)CNA cannot detect solid-like clusters in the melt, we assumed that clusters can be identified by the overlap in phase space between the melt and a pure solid phase. To create such a reference phase space, we first sampled a trajectory of an FCC Cu solid at 1 bar and 400 K at the NPT ensemble using the potential from Mishin et al.[49] and the Nosé-Hoover thermostat and barostat[60,61] implemented in LAMMPS, with damping parameters equivalent to 0.5 and 3.0 ps for the temperature and pressure, respectively. We simulated a $20 \times 20 \times 20$ supercell containing 32,000 Cu atoms. Initial velocities are sampled from a Gaussian distribution scaled to produce the desired temperature, and with zero net linear and angular momentum. The simulation was equilibrated for 40 ps, after which five snapshots separated by 5 ps were saved to create the reference dataset, which contained 160,000 environments.

Using the reference environments, we computed the differential entropy $\delta\mathcal{H}$ of each frame of the solidification trajectory prior to growth. Then, we used a graph theoretical approach to determine the cluster sizes. Specifically, we considered that environments with $\delta\mathcal{H} \leq 0$ with respect to the solid are nodes in a graph, and edges connect environments at most 3.0 Å apart. Then, clusters are defined as 2-connected subgraphs of the larger graph. The cluster sizes are given by the number of nodes in each subgraph, and the maximum cluster in each frame of the trajectory is estimated by the largest subgraph.

## Reporting summary

Further information on research design is available in the Nature Portfolio Reporting Summary linked to this article.

## Data availability

The datasets used for training/testing ML potentials were obtained from the original sources at: • rMD17: https://figshare.com/articles/dataset/Revised_MD17_dataset_rMD17_/12672038 • GAP-20: https://doi.org/10.17863/CAM.54529 • ANI-Al: https://github.com/atomistic-ml/ani-al • TM23: https://doi.org/10.24435/materialscloud:6c-b3. All the data generated in this study, including Supplementary Information, have been deposited in the Zenodo database under the following accession code: https://doi.org/10.5281/zenodo.15025644. Jupyter Notebooks to reproduce the figures of this manuscript are available on GitHub at https://github.com/digital-synthesis-lab/2025-quests-data, with a persistent data storage on Zenodo under the https://doi.org/10.5281/zenodo.15026065.

## Code availability

The code for QUESTS (v2025.03.14) used to perform all the calculations in this manuscript is available on GitHub at the link https://github.com/dskoda/quests. Persistent storage of the code is available on Zenodo under the https://doi.org/10.5281/zenodo.15025957[66].

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

## Acknowledgements

This work was performed under the auspices of the U.S. Department of Energy by Lawrence Livermore National Laboratory (LLNL) under Contract DE-AC52-07NA27344. The authors acknowledge funding from the Laboratory Directed Research and Development (LDRD) Program at LLNL under project tracking codes 22-ERD-055 and 23-SI-006. The authors are grateful to the IAP-UQ group at LLNL for useful discussions, T.H. for providing the data for the denoised copper trajectories, V. Bulatov for the data on the tantalum simulation, and L. Williams for pointing us to the training set for SNAP. D.S.-K. additionally acknowledges support from the UCLA Samueli School of Engineering and from the U.S. Department of Energy, Office of Science, Office of Basic Energy Sciences under Award Number DE-SC0025642. Manuscript released as `LLNL-JRNL-862887`.

## Author contributions

D.S.-K.: Conceptualization; Data Curation; Formal Analysis; Investigation; Methodology; Project Administration; Software; Validation; Visualization; Writing—Original Draft; Writing—Review & Editing; Funding Acquisition; Supervision. S.H.: Data Curation; Investigation; Software; Writing—Review & Editing. B.S.: Data Curation; Investigation; Writing—Review & Editing. F.Z.: Validation; Data Curation; Writing—Review & Editing; Supervision. V.L.: Conceptualization; Data Curation; Writing—Review & Editing; Funding Acquisition; Project Administration; Supervision.

## Competing interests

The authors declare no competing interests.
