## [Transparent Peer Review file · Nature Communications]

Model-free estimation of completeness, uncertainties, and outliers in atomistic machine learning using information theory

Corresponding Author: Professor Daniel Schwalbe-Koda

Version 0:

Reviewer comments:

Reviewer #1

(Remarks to the Author)

The manuscript presents a study of the use of measures of information entropy in feature space in various tasks related to atomistic simulations. The paper is interesting and very thorough. I believe that with some modifications, it would be a valuable contribution to the field.

My main overall concern with the paper is that the exact scope of the claims that are made is not totally clear. For example, the abstract and the title suggests “a unifying theory for data-driven atomistic modeling”, which appears to me to be an overstatement. Instead, the authors provide a compelling and thorough exploration of the idea that the proposed estimator of the information entropy is a useful and computationally efficient surrogate to the ordinary thermodynamic entropy and that it can productively used as a computational tool for a range of purposes. I believe that it would be beneficial to make the aims and claims of the paper clearer and more explicit. E.g., to pick just one example, the paper claims that “Although differences in transition temperatures suggest that the accuracy of our method can be further improved, this quantitative agreement between descriptor distributions, information entropy, and statistical mechanics can simplify computations in first-principles thermodynamics.” It is unclear to me how this statement should be interpreted. Knowing the level of rigor that the statistical mechanics/thermodynamics community is held to, it is unlikely that this method would be used as an alternative to conventional method for high precision calculations unless the relationship between information and thermodynamic entropy is better understood. Can the authors make this statement, and others of the same nature, more precise and concrete?

Second, the empirical relationship between the thermodynamic and information entropies is repeatedly qualitatively and quantitatively touted, but the question of the formal relationship between the two object is surprisingly not explored, which I feel contributes to the “fuzziness” of the claims in the paper. Although an in-depth analysis of the discrepancy for a given set of features is likely to be very challenging, at first sight, it appears rather straightforward to derive conditions for both to be equal. Given its central importance in the paper, this question should be explored and discussed. Can the apparent surprisingly good agreement be rationalized? When can this agreement expected to be especially good or bad?

Supplementary Section 6 discusses a calibration procedure which was applied to essentially “match” information and thermodynamic entropy for an Einstein solid, a procedure which is presumably critical. Is this mapping “universal” or does it need to be recalibrated for each system? E.g., does it depend on the value of the spring constant used to model the Einstein solid? Was the same calibration applied to all examples, or are these recalibrated separately? If so, how does this recalibration affect the predictive nature of the method?

The usefulness of Table 1 is unclear to me, as it compares computational and experimental results. Unless the typical error in exact computational predictions is available, it is unclear to me what message should be taken away from this comparison. Regarding Fig. 2g, it is also unclear to me what this agreement means, given that the CNT line is not a prediction, but a fit to the data. Can the authors provide indications that a good fit would not have been expected unless the cluster size definition is especially physically insightful? The paper often feels like some quantitative comparisons are perhaps “stretched” in a way that would not be necessary if the scope of the paper was more pragmatically defined.

In Section E, the authors claim that their method “is guaranteed to provide a robust uncertainty estimate as it does not rely on

the randomness associated with model training or inference". This statement is unclear: why does the fact that the method is model-free provides guarantees on the quality of the UQ? Shouldn't model-based UQ methods be advantageous in many situations?

(Remarks on code availability)

Reviewer #2

(Remarks to the Author)

As far as I understand, the very basic premise of this paper is rather questionable.

Why? because from the very beginning the authors do not seem to distinguish between the thermodynamic entropy and a part of it, the configurational entropy.

The thermodynamic entropy in its classical mechanics limit, the limit used by the authors, is the entropy of the distribution in classical phase space. This is not the quantity that is given in equation (5).

As a result I cannot agree with key statements of the authors. For example, on page 6, red color in the original, we hypothesize that non-parametric descriptor distributions derived from atomistic simulations can be used to predict thermodynamic entropy differences. Sorry, no.

Of course, there can be limiting situations where most of the thermodynamic entropy differences are configurational in origin. But these are special., E.g., low temperatures. A theory that only applied to such limited applications does not merit the hype and the red highlighting.

Clearly I am unable to recommend publication of this ms.

(Remarks on code availability)

Reviewer #3

(Remarks to the Author)

In their manuscript, the authors consider using information entropy based on atom-centered environments for the study of various phenomena (kinetic events and materials thermodynamics) and to assess the quality of datasets for machine learning tasks. They demonstrate the proposed framework with an extensive and broad set of examples. Derivations and code are provided as supplementary material and in a repository, respectively. As far as I can tell, the datasets used for the examples are not public.

The topic and the presented results are certainly interesting and timely. Utilizing the information content of atom-centered environments, which are central to many machine learning applications, provides an interesting perspective and will probably be appreciated by researchers working in this field. However, in my view, the significance of the results is reduced by some limitations of the proposal

1. I understand that the descriptors introduced in Eqs. (1)-(3) are motivated by earlier studies that investigated their suitability for certain ML applications. Nevertheless, some benchmarking/comparison against other (established) descriptors would certainly be helpful and interesting. In particular, since the authors claim to present a "unifying new theory" it seems appropriate to verify that other descriptors yield comparable results (of course within the computational limits).

2. The resubstitution estimate of the entropy given by Eq. (5) is just an estimate of the true entropy (minor comment: n is not properly defined in the context of the equation). Moreover, using the kernel-density estimate of the probability distribution is known to be problematic for high-dimensional cases (see Ref. 34 or recently in Chaves et al, Entropy 2024, 26(5): 387, doi: 10.3390/e26050387). Therefore, the "quality" of the estimate should be discussed in this context. Of course, Eq. (5) can also be used as a definition, but then it would be better to consistently refer to it as an "estimate of the information entropy". It should also be noted that there are other possibilities to do the estimate (see the references above).

3. The first examples in the manuscript provide some circumstantial evidence for an "atomistic information theory", but there is no real theory. This makes it hard to assess the applicability and generality of the proposed connection between information and thermodynamic entropy, i.e. does it depend on the choice of the kernel, the descriptor, the target system etc. The examples in sections D to F provide a clearer picture, but there are still some questions regarding generality and transferability (for example, when using other descriptors).

Overall, I think that the manuscript in its current form is more suitable for a journal focused on cheminformatics or materials informatics. The concepts are interesting, but in my view, they do not (yet) sum up to a "unifying new theory". This limits the potential impact and makes it less interesting for a broader audience.

Additionally, I have the following questions/comments:

A. The concept of information entropy in the context of atomistic environments is not entirely new and goes back to the 50s (Rashevsky, The Bulletin of Mathematical Biophysics 1955, 17, 229–235). Initially, most works were concerned with topological information as the discussion was based on (molecular) graphs, but recently it was shown how entropy can be defined based on the similarity of atomistic environments, for example, using SOAP similarity (Croy, ACS Omega 2024, 9(18): 20616-20622, doi: 10.1021/acsomega.4c02770). The focus of those works is certainly different from the topic of the manuscript, but they demonstrate that information entropy already plays a role in various questions in molecular/material science.

B. Although there is a section on bandwidth selection in the Supplement, the choice of bandwidth for each case is not further explained. For example, was there some tuning involved and how does the value compare to any “standard” estimate (e.g., Silverman’s)? In general, there is a lot of literature on density estimates which might help optimize the estimates.

C. The overlap of environments is mentioned in Section E. It would be helpful to give the definition already there, as it is currently “hidden” in the Methods section.

D. The similarity of environments (measured by the kernel function) will also depend on the size of the environment (here given by the # of neighbors and the cutoff). How were the values for k and r_c chosen and is there a qualitative impact on the results for different values?

(Remarks on code availability)

I only had a quick look into the code and it seems to be very well structured and documented. I did not try to install it. As far as I can tell, there are no tests or examples provided.

Version 1:

Reviewer comments:

Reviewer #1

(Remarks to the Author)

The new version of the manuscript has a sharper focus which greatly improves the presentation. It makes a convincing case that information entropy is a tool that should be included in the atomistic toolkit and that it offers promising characteristics. As such, the paper could be a useful contribution to the literature. However, I believe that authors still overstate some points, especially those that attempt at quantitative statements, and that these should be tempered before publication.

For example, in Section 2.3 the authors write that “Rather, it provides a conservative, yet deterministic way to compute uncertainties by calculating the relative information content of a data point using the training set as a reference, while making no assumptions on the extrapolation power of the model that was trained on these data points. Therefore, our UQ method can be useful to provide rigorous bounds to epistemic uncertainty regimes,” As far as I can tell, no method to predict uncertainties or confidence intervals is actually presented. Rather, an empirical correlation of an overlap measure between test and train sets and force errors is given. While this is an interesting observation, this falls short of a quantification of errors, at best is it a “qualification of errors” (can the test errors be expected to be large or small?). I doubt that a true predictive algorithm would be possible, given the “model-free” nature of the approach. Certainly, different ML models trained and tested to the same data would perform differently, which the approach presented here has no way of capturing. In fact, Fig 5c shows a counterexample where the force errors of SNAP are uncorrelated with the differential entropy. In that respect, section 2.4 is more balanced, as it shows that information entropy can explain trends, without predictively quantifying them.

The conclusion states that “this work proposes a rigorous way to optimize their training process, automate evaluation of information contents in datasets, and assess the performance of the models against well-defined theoretical bounds.” Again, the paper clearly shows indicative correlations between information entropy and the behavior of ML potentials that are very helpful to rationalize trends, but the paper does not propose rigorous algorithms to “optimize the training process” or assess their performance “against well-defined theoretical bounds”. In that respect, I find that the abstract and introduction reflect the contributions of the paper more appropriately.

Minor point: Similarly, in Section 2.5, the comparison of the critical size from the MD with CNT predictions is a stretch, as it uses a mixture of physical quantities from the potential and from experiments. Compounding these approximations with that inherent to CNT, it is unclear that this direct comparison really brings any credence to the method in comparison with a more conventional CNA-based nuclei identification.

(Remarks on code availability)

The code is provided with examples of use and easy means of installation. The functionalities are straightforward and the level of detail in the readme is appropriate.

Reviewer #3

(Remarks to the Author)

The revisions and replies of the authors have not changed my initial evaluation of the manuscript. Although the new version is more focused, I still don't know what the primary purpose of the article is. From the (at times quite wordy) replies, I understand that they justify their particular choices of the entropy estimation, the descriptors, etc a posteriori by the apparent "utility" of the method for multiple applications in atomistic simulations. This is fine, but it narrows the scope and, in my opinion, makes this work less interesting for researchers outside of the addressed part of the MLIPs community.

In any case, I think that the limitations resulting from some of their choices are not adequately discussed, and some statements are overly broad. For example, in the introduction, they write, "In contrast with other works connecting information entropy and atomistic data, this work proposes a framework to match the assumptions of information theory, ..." (no references given) To be clear: the entropy estimate (Eq. (5)) is well established in information theory, the similarity kernels of atomistic environments have been used and discussed in many works and have been used to calculate information entropy (even if there might be limitations). If the purpose of the article is to show that their implementation of Eq. (5) (and its ingredients) is useful for "model-free quantification ...", they should write that. If they claim to have a new framework (or theory), they should clearly describe what is new and what is known.

As I wrote before, I think the article is well-written and the results are interesting. I don't share the optimistic interpretation of the importance of the "framework", but it might be useful for some researchers. The usefulness of the framework depends in my view crucially on the ability to go beyond the current assumptions, something the authors prefer to investigate in a later publication. Overall, I still think that the manuscript in its current form is more suitable for a journal focused on cheminformatics or materials informatics.

(Remarks on code availability)

Version 2:

Reviewer comments:

Reviewer #1

(Remarks to the Author)

(Remarks on code availability)

Reviewer #3

(Remarks to the Author)

I cannot say that I am convinced by the reply, but the revisions address my comments to some extent. I hope that the authors keep in mind that non-parametric entropy estimation is not unproblematic (especially when kernel density estimates are used) and should be adopted and checked for each data set.

(Remarks on code availability)

AUTHORS' RESPONSE TO THE REVIEWERS

We thank the reviewers for their time and assessment of our work. Since the last manuscript version, we substantially improved the manuscript to address the comments from the reviewers. In particular, the following main improvements were made to the text:

1. The scope of the manuscript was strengthened to make the contributions more clear, especially within the space of machine learning interatomic potentials (MLIPs). The numerous advances provided by our methods — ranging from dataset analysis, explaining trends in errors and heuristics in MLIPs, model free uncertainty quantification (UQ), and outlier detection — are now better emphasized and described throughout the manuscript.
2. A new section was added to showcase the generality of the method for other systems and at larger sizes. As a demonstration on how information entropy and model-free UQ can be used to explain results from new datasets, we analyzed the trends in the newly-proposed TM23 dataset from the Kozinsky group.¹ The original analysis for this dataset explained error trends across transition metals based on electronic structure effects.¹ Here, we show how these effects impact the phase space of the metals and can be alternatively captured in a simple, yet elegant way by our information theoretical analysis. Many other results that resolve extrapolation trends — previously unexplained in the original paper and literature — are also discussed in the manuscript. The additional section demonstrates the usefulness of the method beyond the simpler examples previously explained and helps describe how our method can be of interest towards a broader community, as requested by two reviewers.
3. We consolidated two sections (out-of-equilibrium molecular dynamics (MD) trajectory and identifying errors in MLIP-driven production simulations) into an “outlier detection” section. The similarity between the tasks justifies the use of two production simulations as useful examples on model-free UQ and extraction of physical insights from data. This helps narrow the focus of the study into the data aspects of atomistic simulations. Whereas a number of additional questions can be asked on the topic of nucleation or strength prediction from the physics perspective, we defer these ques-

tions to a future, specialized study where the specifics of the physical mechanisms can be discussed.

4. Similarly, we considered the reviewers' comments on the thermodynamic analysis, and moved the section to the Supplementary Text for a more detailed discussion. We believe that reporting this interesting observation at this stage is crucial to spur further development in the field, but we agree that an incomplete presentation may only serve to defocus the current study, and so we leave follow-up derivations for upcoming work, where we can devote adequate space to a thorough and rigorous exposition. (For instance, the mathematical connections between thermodynamic entropy differences and information are well-known, but the degree that descriptor distributions can be used to approximate certain entropic quantities will be analyzed in detail in the future work.) We believe the present study can be extended in this direction, but we also recognize that a narrower focus of this work on the information aspects is more coherent, per the reviewers comments. We believe the present manuscript is stronger refocused this way. Given the current length of the manuscript, shifting this thermodynamic discussion—which may have a more specialized audience—to future work also allowed us to add important details on the information aspects, particularly related to the points above in the field of MLIPs.
5. Numerous new discussions in the Supporting Text, including additional figures, explanations, and more, were added. The main text was also rewritten (and renumbered) to support the observations above. The main figures were simplified to help the reader with the clarity and focus of the manuscript.

Below, we provide a point-by-point response to the comments. The reviewer comments are in blue text, while our responses are written in black text.

REVIEWER 1

Reviewer: The manuscript presents a study of the use of measures of information entropy in feature space in various tasks related to atomistic simulations. The paper is interesting and very thorough. I believe that with some modifications, it would be a valuable contribution to the field.

Authors: We thank the reviewer for the positive assessment of our manuscript. In the revised version, we provide modifications that address the reviewer’s concern about the scope and ideas of the work. Below, we provide a point-by-point response to the specific questions and concerns.

1. My main overall concern with the paper is that the exact scope of the claims that are made is not totally clear. For example, the abstract and the title suggests “a unifying theory for data-driven atomistic modeling”, which appears to me to be an over-statement. Instead, the authors provide a compelling and thorough exploration of the idea that the proposed estimator of the information entropy is a useful and computationally efficient surrogate to the ordinary thermodynamic entropy and that it can productively used as a computational tool for a range of purposes. I believe that it would be beneficial to make the aims and claims of the paper clearer and more explicit.

Authors: We thank the reviewer for pointing out that this point was unclear and the apparent mismatch between the reading and the takeaways. We have revised the manuscript, removing the confusing statement on the “unifying theory” and rather focusing on the multiple results that illustrate the usefulness of information theory into multiple aspects of atomistic simulations. Accordingly, the manuscript title was changed to: “*Model-free quantification of completeness, uncertainties, and outliers in atomistic machine learning using information theory.*” Even more than before, the connections between information theory and atomistic machine learning are explored in detail. We added an additional section to the manuscript analyzing the newly-proposed dataset TM23 and rationalizing the trends in model error across chemical spaces and data distributions. The extra example showcases how a model-free approach can enable an *a priori* understanding of MLIPs, thus bypassing the need to train models to make predictions and rely on outcomes. Brief observations of the connections to thermodynamic considerations are included, primarily to highlight such observations that merit further investigation and that will be discussed more thoroughly in a future work, given the lack of space for a complete exposition here. We believe this increased focus helps the reader better understand the context of MLIPs and how our method addresses the bottlenecks in this impactful field of research, following the reviewer’s recommendation, resulting in a stronger manuscript, while still pointing to the deeper connections

to thermodynamic entropy that will be discussed elsewhere.

2. E.g., to pick just one example, the paper claims that “Although differences in transition temperatures suggest that the accuracy of our method can be further improved, this quantitative agreement between descriptor distributions, information entropy, and statistical mechanics can simplify computations in first-principles thermodynamics.” It is unclear to me how this statement should be interpreted. Knowing the level of rigor that the statistical mechanics/thermodynamics community is held to, it is unlikely that this method would be used as an alternative to conventional method for high precision calculations unless the relationship between information and thermodynamic entropy is better understood. Can the authors make this statement, and others of the same nature, more precise and concrete?

Authors: We agree with the reviewer’s concern of the rigor connecting the statistical thermodynamics and the empirical observations in our prior analysis. We understand that these concerns merit an entirely new follow-up work to discuss. Accordingly, we now explain some of these connections in the Supplementary Information as an observation that has yet to be fully explained in future works, as it merits a more thorough and comprehensive examination — indeed, we believe that future discussion will benefit a specialized debate within the statistical thermodynamics community. Nevertheless, to answer the reviewer’s question, one opportunity of reducing the computational cost of free energy calculations is that the accuracy of calculations depends on how well the space is sampled. For example, thermodynamic integration approaches require a very controlled set of parameters (e.g., spring constant, fine grid of λ values, long time steps and large number of atoms) to ensure high convergence between them. This process is expensive even at the classical force field level, and perhaps intractable at the quantum chemical accuracy if large-scale simulations are necessary. Simplifying the process of computing configurational entropies could be useful in a range of applications, particularly if created at the DFT level, which remains an interesting field of study. Regardless, in the revised manuscript, we focus on the utility of information theory to atomistic machine learning, thus avoiding a long discussion on this topic. The discussion originally in the main text and now in the Supplementary Text contains more concrete and precise definitions.

3. Second, the empirical relationship between the thermodynamic and information en-

tries is repeatedly qualitatively and quantitatively touted, but the question of the formal relationship between the two objects is surprisingly not explored, which I feel contributes to the “fuzziness” of the claims in the paper. Although an in-depth analysis of the discrepancy for a given set of features is likely to be very challenging, at first sight, it appears rather straightforward to derive conditions for both to be equal. Given its central importance in the paper, this question should be explored and discussed. Can the apparent surprisingly good agreement be rationalized? When can this agreement be expected to be especially good or bad?

Authors: Once again, given the multiplicity of foci for the work, we propose that this issue is discussed in detail in a follow-up work. We believe that the central aspect of this manuscript — that was thus unclear in the previous version — is the model-free analysis of datasets for MLIPs and other aspects of atomistic simulations. Our original intention was to showcase that such a theory is not disconnected from known concepts from thermodynamics, and, in fact, even agreed under certain assumptions. However, as the reviewer pointed out, exploring the nuances of these assumptions and details would dilute even more the main message of this manuscript. Accordingly, we changed the focus of the manuscript to the dataset analysis part, and propose that the manuscript is re-evaluated under this perspective.

Nevertheless, it is worth emphasizing that the connection between the information and thermodynamic entropies are well-known,² and can be traced back to Shannon’s original work.³ One of the original contributions of our formalism is to use descriptor distributions as surrogate probability distributions to bridge the information theoretical perspective with the thermodynamic one. As mentioned in the introduction of the manuscript, other works required the definition of explicit degrees of freedom or were limited to specific scenarios such as pair distribution functions from liquids.^{4–10} However, in the complete discussion of the mathematical derivation in the Supplementary Information, we show that our approximation is a valid entropy approximation from the information perspective and, due to the transitivity between the two formalisms, also corresponds to a valid approximation in configurational entropy. One difference we highlight is that our measure of entropy lacks an on-site term that is typical in the approximation of thermodynamic entropy from simulations.^{4,5} We made this choice for this paper to avoid adding system-specific, learnable parameters to an approach whose goal was to be model-free. Additional details that are important for

practical application of these methods for thermodynamic calculations in different scenarios and levels of approximation are nuanced and deserve fuller attention. Nevertheless, these extensions will be studied in future work.

4. Supplementary Section 6 discusses a calibration procedure which was applied to essentially “match” information and thermodynamic entropy for an Einstein solid, a procedure which is presumably critical. Is this mapping “universal” or does it need to be recalibrated for each system? E.g., does it depend on the value of the spring constant used to model the Einstein solid? Was the same calibration applied to all examples, or are these recalibrated separately? If so, how does this recalibration affect the predictive nature of the method?

Authors: This discussion, which remains in the Supplementary Text of the manuscript for the sake of clarity, was a “universal” mapping for all systems shown in the paper, so it did not have to be recalibrated, at least for the explored examples. Our preliminary findings showed that the calibration was not sensitive to the value of the spring constant (up to an added constant). Once the initial curve of bandwidth versus volume was established, it remained the same for the entire manuscript and across all examples that had been described (FCC-BCC Cu phase boundaries, $\alpha - \beta$ Sn phase boundaries, Cu nucleation, and others). This description remains in the Supplementary Text, in what is currently Sections A.6. and A.11.2. Nevertheless, in the revised manuscript, the bandwidth is now kept constant across all results, with a constant value of 0.015 \AA^{-1} .

5. The usefulness of Table 1 is unclear to me, as it compares computational and experimental results. Unless the typical error in exact computational predictions is available, it is unclear to me what message should be taken away from this comparison.

Authors: The original goal of Table 1 was to demonstrate that the information entropy differences captured trends not only in solid-state phase transformations, but also in phase changes such as melting. Whereas we tried in the original version of the manuscript to consistently reference the results against thermodynamic integration results, comparing the results against melting entropies provided an example on how to use a separate dataset (in this case, DC3) to compute these values of entropy beyond our own simulations. In a sense, the use of a small-scale dataset to compute entropies that, to an extent, roughly capture the trends in melting entropies is an interesting demonstration. For instance, in

the original manuscript we described about the richness in phase space of liquid silicon and germanium, which are also known to have high (experimental) melting entropies, with a metal-to-insulator transition and other interesting effects. Our information theoretical analysis managed to capture that richness in space, as shown by the higher information entropies obtained from the DC3 dataset. Granted, the actual thermodynamic values of melting entropies are much more complex than just the configuration part, which we had stressed throughout the manuscript, and cannot be captured only by the analysis we had provided. Therefore, the results from Table 1 in the previous version of the manuscript provided an opportunity to discuss additional phase transformations and show qualitative trends derived from there. Given the choices made for this revision, instead of creating a broader discussion on the topic, we decided to remove Table 1 from the manuscript, as the connections between information entropy and richness in phase space remain present. For instance, the comparison between accessible phase spaces and environments is still captured in the manuscript through the new example and section studying the TM23 dataset, which has trajectories below and above the melting temperature and exhibit interesting trends in phase space and chemistries.

6. Regarding Fig. 2g, it is also unclear to me what this agreement means, given that the CNT line is not a prediction, but a fit to the data. Can the authors provide indications that a good fit would not have been expected unless the cluster size definition is especially physically insightful? The paper often feels like some quantitative comparisons are perhaps “stretched” in a way that would not be necessary if the scope of the paper was more pragmatically defined.

Authors: The reviewer is correct in pointing out that the CNT line for the cluster size distribution was obtained through a fit. Often, properties of interest for the CNT (such as volumetric free energy differences or surface tensions) are challenging to compute directly from a simulation, as the cluster size distribution itself is difficult to obtain. For instance, in the manuscript, we had originally reported that the CNA method recovers at most 170 FCC-like environments in the entire simulation cell, as shown in Figure R1a below.

In fact, when the same method used to identify critical nuclei described in the main text of our work is used along a-CNA, we observed that most of the times, the atoms did not have neighboring clusters, but were isolated single FCC-like environments. On the other

FIG. R1. **a**, Total number of FCC environments in the Cu nucleation trajectory up until the nucleation event, as detected by the a-CNA algorithm. The a-CNA algorithm predicts that less than 120 FCC-like environments are present in the simulation box, whereas the expected critical nucleus (i.e., values derived from the experimental parameters) should contain more than 100 atoms. **b**, Number of atoms of the largest FCC cluster detected with a-CNA. Until nucleation, the cluster size remains at values close to 1, suggesting the absence of clusters. **c**, The results above lead to a cluster size distribution that barely has a distribution per se, as most frames contain only single-atom “FCC-like” environment. This figure indicates the cluster size distribution for multiple time steps (brighter colors indicate later frames, reproducing Fig. S26 in the current version of the manuscript).

hand, the estimated critical nucleus alone would account for over 100 atoms, which would be unlikely to be predicted by the CNA. This outcome biases the predictions of clusters in the melt, and thus creates cluster size distributions that are not expected by the CNT, as shown in Figs. R1b,c.

In this case, algorithms such as the (a-)CNA are unable to correctly classify environments in simulations, as has been explored by recent work by some of us.¹¹ In the impossibility of classifying even large clusters, it is unlikely that these methods can produce cluster size distributions and, accordingly, recover macroscopic properties such as interfacial energies.

This discussion addresses the reviewer’s concern on the importance of our method in extracting insights from simulations. Whereas we no longer perform the quantitative comparison with the physical properties for this example on nucleation (and thus chose to move the discussion to the SI for consistency, space, and focus), we believe these results justify how a stronger definition of detecting outliers is important to extract physical insights from trajectories. Accordingly, we continue this discussion in the newly-defined “outlier detection”

section (2.5).

7. In Section E, the authors claim that their method “is guaranteed to provide a robust uncertainty estimate as it does not rely on the randomness associated with model training or inference”. This statement is unclear: why does the fact that the method is model-free provides guarantees on the quality of the UQ? Shouldn’t model-based UQ methods be advantageous in many situations?

Authors: Current developments in UQ for potential energy surfaces often try to distinguish between aleatoric and epistemic uncertainty within a model training. The quantification of an uncertainty is commonplace for models such as Gaussian process regression, but is often challenging to be obtained with neural network (NN)-based MLIPs. Recent benchmarks have been trying to understand how to improve the reliability of UQ in MLIPs,^{12,13} specifically in assessing epistemic uncertainty. However, model-based uncertainties often have their own associated uncertainties, creating a recursive problem when evaluating trust in certain model predictions. For instance, ensembles of NNs are typical ways to quantify uncertainty in predictions, and rely on the fact that models trained on slightly different splits of the training set, the stochasticity of the optimizer, initialized weights etc. will exhibit good agreement within the domain of interest, but disagree on the predictions outside of the training set. This so-called epistemic uncertainty is critical to identify in production simulations, as it determines points where the confidence in the predictions is low for unknown reasons. Model-based uncertainties from techniques such as model ensembles do not necessarily guarantee that the (epistemic) uncertainty will be low in extrapolation regimes. This lack of guarantees can be a problem, as overconfidence in NNs can lead to unphysical simulations or oversight of important phenomena, especially at large length and time scales. Therefore, while numerous works are actively trying to address these shortcomings in the field of deep learning and specifically in MLIPs, most remain as heuristics in the field.

Our approach for UQ builds on a very conservative metric to determine the threshold between interpolation and extrapolation. Specifically, we determine that the threshold to determine if a point \mathbf{Y} is out of domain is given by $\delta\mathcal{H} = 0$. As discussed in the manuscript, this threshold emerges naturally from the derived expression of $\delta\mathcal{H}$,

$$\delta\mathcal{H}(\mathbf{Y}|\{\mathbf{X}_i\}) = -\log \left[\sum_{i=1}^n K_h(\mathbf{Y}, \mathbf{X}_i) \right], \tag{1}$$

in the limiting case where the probability distribution of interest is uniform, and thus $\delta\mathcal{H}(\mathbf{Y}|\{\mathbf{X}\}) = 0$, $\forall \mathbf{Y} \in \{\mathbf{X}\}$, and $\delta\mathcal{H}(\mathbf{Y}|\{\mathbf{X}\}) > 0$, $\forall \mathbf{Y} \notin \{\mathbf{X}\}$. In other cases — i.e., when there is more overlap between the reference data points — we have $\delta\mathcal{H}(\mathbf{Y}|\{\mathbf{X}\}) \leq 0$, $\forall \mathbf{Y} \in \{\mathbf{X}\}$. Because this method makes no assumptions on the model performance in o.o.d. scenarios, it considers that predictions are only trustworthy if they are close to the training set, similarly to Gaussian processes. Furthermore, because the information content $\delta\mathcal{H}(\mathbf{Y}|\{\mathbf{X}\})$ relative to the training set $\{\mathbf{X}\}$ is deterministic, we do not rely on assumptions of distributions of predictions to assign a value of uncertainty to the results. This is why we mentioned, in the manuscript, that our model-free approach is a conservative measure of UQ, and benefits from the ability to compute an epistemic contribution to the uncertainty regardless of the model, calibration thresholds, and so on.

Nevertheless, we agree with the reviewer that model-based UQ can be quite beneficial in some cases. Especially with very large training sets, having models that recognize not only the limits of their “extrapolation power” but the uncertainties associated with the predictions remains the gold-standard of UQ for MLIPs (and many other ML applications). The ability to perform inference and obtain uncertainties at low computational costs (for example, avoiding the larger costs of model ensembles or the potential errors in parametric density estimations) would be ideal in any scenario. As this goal has not been fully achieved yet — and current approaches in the literature cannot offer this desired performance — we are proposing an alternative method that complements future developments in model-based UQ. In fact, a typical problem in this field is that apparently large training sets can actually be quite narrow, due to underestimation of redundancy and compaction in the model space, issues that our methodology allows to systematically analyze. We clarified these point in the main text, first paragraph of current Section 2.3:

While this approach can be expensive for large datasets, it is easily parallelizable and is guaranteed to provide a robust uncertainty estimate, as it does not rely on the randomness associated with model training or inference. Rather, it provides a conservative, yet deterministic way to compute uncertainties by calculating the relative information content of a data point using the training set as a reference, while making no assumptions on the extrapolation power of the model that was trained on these data points.

REVIEWER 2

1. As far as I understand, the very basic premise of this paper is rather questionable. Why? because from the very beginning the authors do not seem to distinguish between the thermodynamic entropy and a part of it, the configurational entropy.

Authors: We respectfully disagree with the Reviewer. While a more detailed discussion could have been made on the topic to ensure maximal clarity and rigor with respect to the conventional definitions, we did distinguish between the thermodynamic entropy and its configurational component in our manuscript, contrary to the reviewer’s main claim. In the original version of the manuscript, we already emphasized that our premise was that the information entropy of descriptor distributions relates to *configurational entropy differences*—among the many other useful correlations presented for MLIPs, dataset analysis, and more. These statements were distributed carefully in the manuscript, especially in (formerly) Section II.B. For instance, the following excerpts in the original manuscript emphasize the point that the discussion was present in the original manuscript (lines numbers relative to the previous version of the manuscript):

- **Line 127:** “*we hypothesize that non-parametric descriptor distributions derived from atomistic simulations can be used to predict **thermodynamic entropy differences**.*”
- **Line 129:** “***Experimental entropy values include numerous additional contributions from configurational (e.g., disorder in solid solution), vibrational (e.g., position and momenta), electronic, magnetic, and other effects not accounted for by our structure-based descriptor approach. Hence, we restrict our comparison to entropy differences obtained from thermodynamic integration (TI) at constant temperature and volume/pressure. This eliminates the dependence of the computed values on the partition function due to momenta of atoms, and still provides a useful way to compute entropy values that otherwise depend on costly simulations.***”
- **Line 155:** “*Nevertheless, despite the approximations from the descriptors and KDE, we successfully recovered not only trends in thermodynamic values, but also the exact values of **entropy differences** for the BCC and FCC Cu.*”

- **Line 633:** *“Moreover, although our model succeeded in predicting relative configurational entropy differences, computation of true entropy values requires incorporating effects of velocity (i.e., complete vibrational entropy), electronic, magnetic, and other components to the final results, all of which influence the phase transformations of materials.”*

The excerpts above suggest that the reviewer may have overlooked this discussion in the manuscript, which properly addressed the distinctions of the actual thermodynamic entropy and the configurational component of it. In fact, we had carefully described these limitations of the information-theoretical method in the main text and Supporting Information. As pointed out to the other reviewers, we understand that the concerns on the perceived claims deserves an additional follow-up manuscript to draw the full details of these connections, acknowledging that the brief exposition in the original version of this manuscript inadvertently added confusion and removed focus from the main points.

Nevertheless, this does not detract from the merits of our current work and the application of the information-theoretical method to multiple other tasks related to atomistic simulations that have not been evaluated by the reviewer. Thermodynamic assumptions aside, in the manuscript we still showcase how the information theoretical approach resolves multiple problems in atomistic ML and simulations in general. With the new section on the TM23 and the extended explanations on the ability to perform atomistic simulations, we explain how phenomena regarded as heuristics in atomistic ML can be traced back to the diversity of points and the distribution of data in an embedding space. This quantitative agreement between model predictions and our information entropy analysis provides a new tool for atomistic simulations and extends a formalism that remained qualitative for this specific field.

We hope the reviewer may revisit this criticism in light of the main points of the (revised) manuscript, which has not backtracked, but rather better focused the overall messaging. This response letter and the reviewed manuscript both showcase extensive evidence of the novelty and merits of our work, which offers strong examples of the fundamentals and application of our method and several novel practical results; however, these aspects were not addressed by this reviewer. We believe the interest and importance of the work remains even when the discussion on the thermodynamic connections is reported solely as an interesting,

surprising observation, in this revised version.

2. The thermodynamic entropy in its classical mechanics limit, the limit used by the authors, is the entropy of the distribution in classical phase space. This is not the quantity that is given in equation (5). As a result I cannot agree with key statements of the authors. For example, on page 6, red color in the original, we hypothesize that non-parametric descriptor distributions derived from atomistic simulations can be used to predict thermodynamic entropy differences. Sorry, no.

Authors: The information and thermodynamic entropies are knowingly not the same quantity, though their connections have been explored for a long time.^{2,3} However, we also did not claim the equivalence of these two quantities in the original paper. As described in the points above, we merely pointed out that the agreement between them was surprising, and that information entropy **differences** correlated with thermodynamic entropy **differences**, in the examples and assumptions under study. These quantities are computed in the context of atomistic simulations, specifically when it comes to some examples of free energy calculations. For instance, numerous *ab initio* phase diagrams have been computed essentially by comparing the configurational component of the thermodynamic entropy (and supplemented by the electronic or magnetic entropies when appropriate) at each (T, P) pair. This approach to the computation of phase diagrams, therefore, understands that the thermodynamic quantities are not the same. Nevertheless, when free energy differences at a constant (T, P) $\Delta G = \Delta H - T\Delta S$ are considered, then thermodynamic **entropy differences** have major contributions in most cases. At a given temperature, the partition function due to the distribution of momenta will be identical for two different phases, rendering this component less important than the configurational one to determine phase stability. Similarly, the contributions of electronic and magnetic entropies may or may not be used in the construction of the phase diagram. Very often, obtaining *ab initio* phase diagrams from atomistic simulations simplifies the framework and does not include these contributions, which is a reasonable assumption in several cases. Obtaining methods that accelerate the computation of these quantities is still relevant for a range of applications in materials thermodynamics.

Therefore, it is straightforward that information entropy is not thermodynamic entropy; this was never the claim from the manuscript. Instead, we reported a surprising observation where the values of information entropy somehow agreed with **thermodynamic entropy**

differences under assumptions that are common in the calculation of phase diagrams. Furthermore, as mentioned in the response to Reviewer 1, the formalism corresponds to a valid entropy term with the exception of the absence of an on-site entropy term. When summarizing these findings in the introduction in the context of a much broader manuscript, however, we did not expand on these definitions, and left them to the main text of the work. As discussed before, we believe that the original manuscript contained enough information to bring up this point. However, this should not be an issue in the revised refocused manuscript, where we report the findings on the correlations between configurational entropy differences and information entropy only as observations worthy of future studies.

3. Of course, there can be limiting situations where most of the thermodynamic entropy differences are configurational in origin. But these are special, E.g., low temperatures. A theory that only applied to such limited applications does not merit the hype and the red highlighting.

Authors: Once again, we believe this discussion was present in the main text, and disagree with the reviewer’s assessment. First, we agree that there are several different conditions where thermodynamic entropy differences are mostly configurational, especially within solid-state phase transformations. However, one of the examples shown in the original manuscript is the FCC-BCC phase boundary of copper at higher temperatures and pressures. Whereas “high” and “low” temperature is arguably difficult to define in a general, domain-agnostic manner, we argue that the reviewer’s comment does not necessarily agree with what we originally showed. The phase diagram of Cu was computed between 3000 to 5000 K and 180 to 280 GPa, and one can argue this cannot be immediately dismissed as a “low-temperature limiting scenario.” In practice, numerous efforts focus on computing phase diagrams of elemental structures at higher temperatures and pressures using methods similar to the ones we described in the response above.

We also argue that, even if this agreement was valid only in limiting (and relevant) scenarios, it would still be a surprising result from an atomistic simulation perspective. The connection between atom-centered representations and potential energy surfaces is well-known, since the proposed strategy from Behler and Parrinello,¹⁴ but few similar examples have been proposed even for limiting scenarios of configurational entropy. As we believe this will spur further investigation on the topic, we now report this observation in the Supporting

Text with reduced emphasis. At the same time, we argue in this letter — and both in the original and revised manuscripts — that the thermodynamic entropy is not the entirety of our findings, let alone the main finding of the work. In fact, the contributions of our information-theoretical formalism for atomistic simulations is quite diverse, and we revised the manuscript to focus on the range of phenomena related to atomistic ML, as suggested by the other reviewers. Currently, we believe that the focused version stands even stronger than the previous version of the manuscript, yet does not alienate the previously reported information from the reader.

REVIEWER 3

Reviewer: In their manuscript, the authors consider using information entropy based on atom-centered environments for the study of various phenomena (kinetic events and materials thermodynamics) and to assess the quality of datasets for machine learning tasks. They demonstrate the proposed framework with an extensive and broad set of examples. Derivations and code are provided as supplementary material and in a repository, respectively. As far as I can tell, the datasets used for the examples are not public.

The topic and the presented results are certainly interesting and timely. Utilizing the information content of atom-centered environments, which are central to many machine learning applications, provides an interesting perspective and will probably be appreciated by researchers working in this field. However, in my view, the significance of the results is reduced by some limitations of the proposal

Authors: We thank the reviewer for the positive evaluation of our work. In this letter, we address the reviewer’s concerns regarding the limitations of our approach and the significance of the results. The manuscript was thoroughly reviewed to emphasize the advances, particularly on the information-theoretical analysis of MLIPs and their impact in atomistic simulations. We also addressed the reviewer’s concerns in the main manuscript, and provide a detailed discussion here.

We emphasize that most of the datasets used in this work are public and are well-known benchmarks in the field. We have clarified this point on the Data Availability section. The non-public datasets (i.e., copper nucleation trajectories and tantalum snapshot shown in

Fig. 5 of the revised manuscript) will be made public at publication time.

1. I understand that the descriptors introduced in Eqs. (1)-(3) are motivated by earlier studies that investigated their suitability for certain ML applications. Nevertheless, some benchmarking/comparison against other (established) descriptors would certainly be helpful and interesting. In particular, since the authors claim to present a “unifying new theory” it seems appropriate to verify that other descriptors yield comparable results (of course within the computational limits).

Authors: We thank the reviewer for the suggestion regarding benchmarking. In our manuscript (and the extensive discussion on the Supplementary Information), we demonstrate how the approach is provably generalizable to any representation \mathbf{X} of atomic environments, provided that the following is kept in mind:

- **Completeness:** the values of entropy depend on the ability of the descriptor to represent atomistic environments in a complete manner. For example, SOAP descriptors¹⁵ have been demonstrated to be incomplete representations,¹⁶ meaning that some different atomic environments map to the same representation. This necessarily reduces the entropy of the system, given that overlapping points always reduce the entropy of a dataset. This conclusion does not necessarily require testing to be understood, given that the mathematical formalism is well-defined.
- **Metric space:** each descriptor has a different metric space, which first must be well-defined mathematically; then, the choices of distances (and kernels) may influence the final results, sometimes in ways that can be incomparable for certain choices of descriptor. For instance, for an arbitrary representation, it is not obvious whether continuous variations on the structural space may lead to continuous distances in the descriptor space.¹⁷ This influences our ability to map distributions of atomic environment back into continuous spaces that correctly represent their information. We have constructed our descriptor to avoid (or minimize) these singularities, circumventing such uncertainties.
- **Bandwidth:** the choice of the bandwidth is descriptor-dependent, and thus will require a separate analysis for each (making some comparisons invalid).

Given the caveats above, one can still perform the analysis of a generic dataset using given descriptors following the method described in the manuscript. To illustrate this point, following the approach described in the manuscript (Methods and Supp. Text A.6), we computed the SOAP distance of environments in two FCC structures as a function of the strain, obtaining the results shown in Fig. R2 of this letter. In this example, the asymmetry of the SOAP descriptors under tension instead of compression illustrates how metric spaces can be different depending on the choice of descriptor. This difference may lead to different results of the entropy of a dataset, considering that structures under tension are more similar to each other compared to structures under compression. This proof of concept demonstrates, therefore, that extending the analysis to another descriptor is not as immediate as it seems, and may require recomputing all the results in the manuscript under multiple assumptions.

FIG. R2. Euclidean distance between SOAP descriptors ($n = 6, r_c = 5 \text{ \AA}, \ell = 8$) of ideal FCC structures with rescaled lattice parameters (strain). This parallels the results from Fig. S6 from the Supplementary Information, though a comprehensive exploration of the effects of SOAP descriptors and information entropy has yet to be analyzed.

Importantly, we wanted to avoid casting this work as a benchmark between descriptors for two reasons. First, the focus of the manuscript is to demonstrate the utility of information entropy on multiple applications in atomistic simulations. Demonstrating the transferability of each application while also using caveats (e.g., different metrics, bandwidths, and so on) on the different descriptors would make the text confusing to the reader while adding little information — i.e., that it works beyond the proposed QUESTS descriptor. Second, and

more critically, not all analyses would be possible with arbitrary descriptors. We invested substantial time in parallelizing and optimizing the code for QUESTS to enable the analysis of large supercells and fast computation of kernel density estimates. At this stage, using different descriptors would require either optimizing additional code for others or accepting large computational costs for the calculation of the entropy. Therefore, while we understand the reviewer’s suggestion of computing different descriptors, we believe such computation can be more adequate for future work focusing on different representations rather than this particular work. This is especially the case with our presently revised manuscript, where we have removed the confusing comments on a “unifying theory.”

2. The resubstitution estimate of the entropy given by Eq. (5) is just an estimate of the true entropy (minor comment: n is not properly defined in the context of the equation). Moreover, using the kernel-density estimate of the probability distribution is known to be problematic for high-dimensional cases (see Ref. 34 or recently in Chaves et al, Entropy 2024, 26(5): 387, doi: 10.3390/e26050387). Therefore, the “quality” of the estimate should be discussed in this context. Of course, Eq. (5) can also be used as a definition, but then it would be better to consistently refer to it as an “estimate of the information entropy”. It should also be noted that there are other possibilities to do the estimate (see the references above).

Authors: The reviewer is correct in saying that Eq. (5) is an estimate of the true entropy, though it is also accurate to say that all works dealing with real, non-analytical toy examples are only able to provide estimates of the true entropy. The integral over the support of the distribution in Eq. (4) is almost always intractable unless the probability distribution is known beforehand. This point had already been emphasized in the main text and Supplementary Information by pointing out that Eq. (5) represents a non-parametric estimate rather than the true entropy. Throughout the manuscript, we avoided referring to the information entropy measure as the “true entropy” with respect to the underlying probability distribution. On the other hand, we believe that carrying out the definition of “estimate of the information entropy” at every occurrence would also burden the manuscript with the term. Therefore, we added in the main text a note on this topic, and how we opt for the “entropy” word as defined by Eq. (5) instead of carrying the longer term.

We emphasize that, despite the approximations, our manuscript demonstrates that the

empirical entropy remains useful in a range of applications, very much like the empirical risk minimization is extremely useful in deep learning even if an incomplete approximation. Future work can focus on studying the effects of approximation of the true entropy, though much work on the subject has already been done in the field, as pointed out by the reviewer. In the main text, we had cited an important work on the subject,¹⁸ and added one extra example suggested by the reviewer.

Regarding the quality of the estimate, we believe the entirety of the manuscript demonstrates the utility of this approximation. We agree that more mathematical rigor could open new directions connecting information theory and atomistic simulations, but we also believe that defining the “quality” of a distribution is still unclear at this stage. In our perspective, we propose to use this manuscript to describe a tool with large utility in the field of MLIPs, as it explains a range of results even if its inner workings can be still improved with further studies. Future mathematical derivations can follow this empirical thread and resolve some of the issues discussed in this letter and in the Discussion section of our manuscript. Along with this, we believe that being able to define what it means to have a distribution of “good quality” in the context of atomistic simulations is a reasonable outcome of future investigations. However, we understand the current manuscript already contains a very high density of content, including some on the mathematical perspective, and adding additional demonstrations on the nuances of the approximation would fall beyond the scope of this work. Despite that, we added a short discussion on this topic in the main text, enabled by the present refocusing of the manuscript, to clarify that these topics and the mathematical nuances can be analyzed in more detail in subsequent works.

3. The first examples in the manuscript provide some circumstantial evidence for an “atomistic information theory”, but there is no real theory. This makes it hard to assess the applicability and generality of the proposed connection between information and thermodynamic entropy, i.e. does it depend on the choice of the kernel, the descriptor, the target system etc. The examples in sections D to F provide a clearer picture, but there are still some questions regarding generality and transferability (for example, when using other descriptors).

Authors: In alignment with comments from other reviewers, we decided to improve the quality of the manuscript by increasing the focus on the MLIP, UQ, and dataset analysis, and

mention the connections with thermodynamics in passing. We agree that the importance of these distinctions merits a separate paper with these results for clarity, as illustrated by the comments from Reviewer 2. Therefore, by removing the focus on these issues (and the generality according to other approaches within non-parametric estimates and kernels), we believe the work still provides significant insight and utility to the field of atomistic simulations and machine learning for materials/chemical sciences. As such, we expanded on the topics previously labeled D to F (now 2.2 to 2.5) to clarify even more the issues of transferability and generality. For instance, we applied our approach to a completely new dataset containing simulations of 27 transition elements (TM23). The analysis of the trends in errors and chemistries of these elements illustrates how the method can be useful beyond the initial examples proposed in the manuscript. We believe these results strongly support the generality of our method in a way that could not be achieved by understanding whether different kernels lead to the same result. Large amounts of works have been performed in the statistics literature to compare kernels, and while the question is interesting, we prefer to first focus on the many results discussed in this manuscript before adding one extra degree of freedom that is fairly well-studied. We believe that if the work had focused on the comparison between kernels and approaches to optimize a KDE, its impact would have been diminished compared to the quantitative results highlighted in the manuscript. Furthermore, even the supplementary discussion already provides extensive evidence on the use of information theory for atomistic simulations, leaving space for the empirical analysis of KDEs for other studies.

Regarding the lack of clarity in the atomistic information theory, we have now refined the manuscript to convey this message in a more clear way, and appreciate the comments of the reviewers in pointing this out. The message is clearer now, with the focus on the model-free analysis of dataset completeness, uncertainty quantification, and outliers in atomistic machine learning being possible due to the formulation of a framework that bridges principles from information theory into the context of atomistic simulations.

4. Overall, I think that the manuscript in its current form is more suitable for a journal focused on cheminformatics or materials informatics. The concepts are interesting, but in my view, they do not (yet) sum up to a “unifying new theory”. This limits the potential

impact and makes it less interesting for a broader audience.

Authors: We thank the reviewer once again for the positive evaluation of our work. At the request of the reviewers, we removed references to a “unifying new theory,” but emphasize the generality of the method for a broad community of atomistic simulations. Throughout the revised manuscript and this response letter, we emphasized that the largest contribution of our work is in explaining results in atomistic machine learning that, so far, remained as heuristics in the field. Though several works have analyzed the intersection of information theory and the chemical/materials sciences (see point 5 below), we are unaware of a theoretical approach that can be used to explain trends in dataset errors, perform uncertainty quantification analysis, evaluate completeness and diversity of datasets, detect outliers in production simulations, and much more. Most publications in the field work with empirical evidences of uncertainty (e.g., distribution of predictions from ensembles) and model outcomes. In this context, the rationale behind different model performances in different datasets remained heuristic, with most approaches analyzing model errors in a post-hoc manner instead of an *a priori* one. For instance, most work in the field opts for training models to assess whether extrapolation is possible. Our current approach provides a theoretical framework to avoid these post-hoc explanations and focus on understanding data distributions to improve atomistic simulations.

In addition, we explored numerous other topics in the manuscript that derive from the ideas above, among which is the discussion on the thermodynamics and connections with information entropy. For the sake of brevity, however, we decided to move the analysis on the thermodynamics and kinetic events to the Supplementary Information, while also reducing the emphasis of the manuscript on these results to allow further works to better explore them with the necessary rigor. Nevertheless, the overall manuscript and additional information (including a complete new section demonstrating the generality of our method to 27 different elements and the ensuing trends) provide an improved picture of the potential impact of this work towards the field of atomistic simulations. Furthermore, this journal has a history of publishing works that describe advances in developing and applying MLIPs for molecular and materials science (see Refs. 19–25 for a few recent examples from different groups), with high impact and without loss of generality for a broader community. Thus, given the combination of generality, usefulness, and ability to explain a range of results in

atomistic simulations and ML-accelerated atomistic simulations, we believe this journal is appropriate for our manuscript, which can be of high interest to a broad audience.

Reviewer: Additionally, I have the following questions/comments:

5. The concept of information entropy in the context of atomistic environments is not entirely new and goes back to the 50s (Rashevsky, *The Bulletin of Mathematical Biophysics* 1955, 17, 229–235). Initially, most works were concerned with topological information as the discussion was based on (molecular) graphs, but recently it was shown how entropy can be defined based on the similarity of atomistic environments, for example, using SOAP similarity (Croy, *ACS Omega* 2024, 9(18): 20616–20622, doi: 10.1021/acsomega.4c02770). The focus of those works is certainly different from the topic of the manuscript, but they demonstrate that information entropy already plays a role in various questions in molecular/material science.

Authors: We thank the reviewer for the provided literature, which we have added to our manuscript to enrich the discussion. In our initial introduction, we already acknowledged the history of this field, which has been fruitful for almost 80 years. Other works, including the ones from Rashevsky or Croy mentioned by the reviewer, demonstrate that information entropy provides a useful interpretation for molecular similarity at the *molecular graph* perspective. The main innovation of our work is showcasing how this concept can be generalized towards atomistic environments from *potential energy surfaces*, which permeate numerous problems in atomistic simulations, including dataset analysis, uncertainty quantification, outlier detection, explaining trends in errors from MLIPs, and more. These results are possible because of the careful derivation of the information entropy based on atom-centered representations, and the strong connections between our formalism and the original information theory proposed by Shannon. In our manuscript and Supplementary Text, we describe how it differs, for example, from excellent works such as the ones from Oganov and Valle²⁶ and Perez *et al.*^{27,28} These works provide a measure of entropy that relates to information theory, but does not recover some of the main results such as well-defined bounds, expected behavior in the limiting cases of degenerate distributions, and so on. Our extended discussion in the Supplementary Text demonstrates how this is crucial for many of the results in our paper. For instance, a measure of entropy that does not satisfy the results in Supp. Text A.1 – A.3 may be unable to compute information gaps, predict errors

or uncertainties, or evaluate the diversity of a dataset. If entropies were computed with the formalism proposed in Refs. 27,28, it would not be possible to explain datasets that sample redundant information, as two degenerate environments would lead to a singularity in the entropy expression. From the PES perspective — which more often than not favors sampling a local energy minimum multiple times according to the Boltzmann distribution — this is an unphysical behavior. It is nearly inevitable to end up with identical environments in an atomistic dataset given the nature of the sampling. Therefore, an entropy measure that is not well-posed prevents most computations shown in this manuscript and its usefulness in dataset and model *analysis* (as opposed to dataset *construction*).

Furthermore, the current understanding in the literature relies on model predictions or qualitative visualizations to explain trends in dataset richness. For instance, it is a known practice in the field to plot datasets using dimensionality reduction techniques and use these qualitative visualizations to demonstrate measures of diversity. Whereas this approach is useful, it does not quantify exactly how much diversity or information is contained in a particular dataset. On the other hand, our methods provide a clear, general pathway to **quantify** measures of information in materials and molecular datasets in a way that makes sense and reproduce expected trends in errors. To further demonstrate this point in the revised manuscript, we added an analysis on the newly proposed TM23 dataset from Owen *et al.*,¹ which generated data for 27 different transition metals and obtained trends in model error. In the original paper, the authors relate the model predictions to many-body interactions emerging from the electronic structure of the metals. In our analysis, we show how quantifying the amounts of information in the datasets can be used to provide a complementary explanation that is particularly useful for practitioners in the field. Often, practitioners (i.e., users or developers of MLIPs for certain applications) need accurate explanations on why their models are failing or whether their datasets are sufficient to perform production simulations. This is one of the main contributions of our work, which introduces substantial novelty compared to current works understanding molecular distributions via other information theoretical formalisms.

In the revised manuscript, we provided a more complete discussion on these topics to address this question from the reviewer (Introduction, Sections 2.2 – 2.4).

6. Although there is a section on bandwidth selection in the Supplement, the choice

of bandwidth for each case is not further explained. For example, was there some tuning involved and how does the value compare to any “standard” estimate (e.g., Silverman’s)? In general, there is a lot of literature on density estimates which might help optimize the estimates.

Authors: The choice of the bandwidth is a typical question regarding KDEs, and also describes one concern from Reviewer 1. To elaborate on that, we added more explicit descriptions in the revised manuscript (Methods and Supp. Text A.6). To summarize here, our choice of the (constant) bandwidth in the manuscript was informed by a physical constraint. Essentially, our idea of having a bandwidth is to rescale the metric space of our descriptors to provide meaningful considerations between the difference between two structures. Granted, there is a multitude of ways with which two atomic environments can be considered different. In our case, we computed the distance between two FCC structures differing by a 1% strain, and chose that as the bandwidth (Fig. S6 of the SI). This choice represents a reasonable ballpark measure of dissimilarity of two atomic environments, and we found it generally effective. Nevertheless, by keeping the bandwidth constant at 0.015 \AA^{-1} throughout our manuscript, we obtained numerous (fair) results with our information theoretical analysis, supporting that the choice led to interpretable, useful, and consistent results. The newly added analysis of the TM23 dataset¹ showcases the strengths of our approach: by analyzing the diversity of configurations across phases and elements, we explained the trends in errors from MLIPs in a way that complements the model-based perspective existent in the field.

On the other hand, the Silverman estimate assumes that the density and the kernel are Gaussian, and minimizes the mean integral squared error. However, this choice of bandwidth typically relies on the observations, which can be quite system-dependent in our case. To illustrate this with a toy example, the potential energy surface of a double-well potential could generate a data distribution that is bimodal, which would cause the variance of the data observations to increase and oversmooth the actual distribution.²⁹ This is the opposite of what is intended with our project: with a single choice of bandwidth that is also independent of the data distribution (for the information side), we can assess the coverage of the phase space in datasets for MLIPs. Therefore, our method provides a strategy that is generalizable and does not rely on optimizing an estimate. In fact, it is still unclear what would one describe as an “optimal estimate” in the space of atom-centered representations,

as the high dimensionality of these distributions could skew the interpretation in ways that separate them from the physical reality of interest. Therefore, we recognize that the extensive literature in the field for optimizing non-parametric estimates is extremely useful, yet not directly applicable to the problem at hand.

7. The overlap of environments is mentioned in Section E. It would be helpful to give the definition already there, as it is currently “hidden” in the Methods section.

Authors: We added the definition of overlap to the referred section (previously Section E, current 2.3).

8. The similarity of environments (measured by the kernel function) will also depend on the size of the environment (here given by the # of neighbors and the cutoff). How were the values for k and r_c chosen and is there a qualitative impact on the results for different values?

Authors: We chose the values of k and r_c to match typical values used in MLIPs. The difference with respect to use for MLIPs, however, is that density-based representations, symmetry functions, and graph neural networks consider a fixed radius regardless of the density of the system under analysis, and embed the environment into a fixed-length descriptor regardless of the number of atoms. In our case, we used a representation whose length is determined by the number of neighbors. Similar approaches have been successful for unsupervised analysis of molecular and materials datasets,^{17,30} and have the advantage of: (1) being extremely fast to compute; and (2) potentially being extended to reconstruct the environment, following the approach from Widdowson et al.¹⁷ The computation of the QUESTS descriptors also allows for efficient parallelization, as it depends only on neighbor lists, which helps with the generation of descriptors for extremely large systems. For instance, computing the descriptors for the tantalum dataset described in the main text, which has 32.5M atoms, takes less than two minutes using a single node of the Ruby supercomputer at LLNL (56 threads, Intel Xeon CLX-8276L CPUs), which is remarkable given the needs for large-scale analysis and other optimized representations in the field (e.g., Ref. 31). The use of a small, fixed-length descriptor also makes it efficient to perform a kernel density estimate, as computing the kernel matrix requires $\mathcal{O}(kn^2)$ operations, where n is the

number of atoms in the dataset.

9. I only had a quick look into the code and it seems to be very well structured and documented. I did not try to install it. As far as I can tell, there are no tests or examples provided.

Authors: Thank you for taking the time to verify the code. We added additional datasets and examples to the repository to demonstrate how to perform the computations and provide improved documentation for users.

REFERENCES

- ¹C. J. Owen, S. B. Torrisi, Y. Xie, S. Batzner, K. Bystrom, J. Coulter, A. Musaelian, L. Sun, and B. Kozinsky, “Complexity of many-body interactions in transition metals via machine-learned force fields from the TM23 data set,” *npj Computational Materials* **10**, 92 (2024).
- ²E. T. Jaynes, “Information theory and statistical mechanics,” *Physical Review* **106**, 620 (1957).
- ³C. E. Shannon, “A mathematical theory of communication,” *The Bell System Technical Journal* **27**, 379–423 (1948).
- ⁴D. C. Wallace, “Correlation entropy in a classical liquid,” *Physics Letters A* **122**, 418–420 (1987).
- ⁵A. Baranyai and D. J. Evans, “Direct entropy calculation from computer simulation of liquids,” *Physical Review A* **40**, 3817–3822 (1989).
- ⁶J. R. Morris and K. M. Ho, “Calculating Accurate Free Energies of Solids Directly from Simulations,” *Physical Review Letters* **74**, 940–943 (1995).
- ⁷C. D. Van Sicen, “Information entropy of complex structures,” *Physical Review E* **56**, 5211–5215 (1997).
- ⁸R. L. C. Vink and G. T. Barkema, “Configurational Entropy of Network-Forming Materials,” *Physical Review Letters* **89**, 076405 (2002).
- ⁹B. J. Killian, J. Yundenfreund Kravitz, and M. K. Gilson, “Extraction of configurational entropy from molecular simulations via an expansion approximation,” *The Journal of Chemical Physics* **127**, 024107 (2007).

- ¹⁰Y. Huang and M. Widom, “Vibrational Entropy of Crystalline Solids from Covariance of Atomic Displacements,” *Entropy* **24**, 618 (2022).
- ¹¹T. Hsu, B. Sadigh, N. Bertin, C. W. Park, J. Chapman, V. Bulatov, and F. Zhou, “Score-based denoising for atomic structure identification,” *npj Computational Materials* **10**, 155 (2024).
- ¹²Y. Hu, J. Musielewicz, Z. W. Ulissi, and A. J. Medford, “Robust and scalable uncertainty estimation with conformal prediction for machine-learned interatomic potentials,” *Machine Learning: Science and Technology* **3**, 045028 (2022).
- ¹³A. R. Tan, S. Urata, S. Goldman, J. C. Dietschreit, and R. Gómez-Bombarelli, “Single-model uncertainty quantification in neural network potentials does not consistently outperform model ensembles,” *npj Computational Materials* **9**, 225 (2023).
- ¹⁴J. Behler and M. Parrinello, “Generalized Neural-Network Representation of High-Dimensional Potential-Energy Surfaces,” *Physical Review Letters* **98**, 146401 (2007).
- ¹⁵A. P. Bartók, R. Kondor, and G. Csányi, “On representing chemical environments,” *Physical Review B* **87**, 184115 (2013).
- ¹⁶S. N. Pozdnyakov, M. J. Willatt, A. P. Bartók, C. Ortner, G. Csányi, and M. Ceriotti, “Incompleteness of atomic structure representations,” *Physical Review Letters* **125**, 166001 (2020).
- ¹⁷D. Widdowson and V. Kurlin, “Resolving the data ambiguity for periodic crystals,” *Advances in Neural Information Processing Systems (NeurIPS 2022)* **35**, 24625–24638 (2022).
- ¹⁸J. Beirlant, E. J. Dudewicz, L. Györfi, E. C. Van der Meulen, *et al.*, “Nonparametric entropy estimation: An overview,” *International Journal of Mathematical and Statistical Sciences* **6**, 17–39 (1997).
- ¹⁹K. T. Schütt, F. Arbabzadah, S. Chmiela, K. R. Müller, and A. Tkatchenko, “Quantum-chemical insights from deep tensor neural networks,” *Nature Communications* **8**, 13890 (2017).
- ²⁰J. S. Smith, B. Nebgen, N. Mathew, J. Chen, N. Lubbers, L. Burakovsky, S. Tretiak, H. A. Nam, T. Germann, S. Fensin, and K. Barros, “Automated discovery of a robust interatomic potential for aluminum,” *Nature Communications* **12**, 1257 (2021).
- ²¹D. Schwalbe-Koda, A. R. Tan, and R. Gómez-Bombarelli, “Differentiable sampling of molecular geometries with uncertainty-based adversarial attacks,” *Nature Communications* **12**, 5104 (2021).

- ²²S. Batzner, A. Musaelian, L. Sun, M. Geiger, J. P. Mailoa, M. Kornbluth, N. Molinari, T. E. Smidt, and B. Kozinsky, “E(3)-equivariant graph neural networks for data-efficient and accurate interatomic potentials,” *Nature Communications* **13**, 2453 (2022).
- ²³S. Takamoto, C. Shinagawa, D. Motoki, K. Nakago, W. Li, I. Kurata, T. Watanabe, Y. Yayama, H. Iriguchi, Y. Asano, T. Onodera, T. Ishii, T. Kudo, H. Ono, R. Sawada, R. Ishitani, M. Ong, T. Yamaguchi, T. Kataoka, A. Hayashi, N. Charoenphakdee, and T. Ibuka, “Towards universal neural network potential for material discovery applicable to arbitrary combination of 45 elements,” *Nature Communications* **13**, 2991 (2022).
- ²⁴J. Chapman, T. Hsu, X. Chen, T. W. Heo, and B. C. Wood, “Quantifying disorder one atom at a time using an interpretable graph neural network paradigm,” *Nature Communications* **14**, 4030 (2023).
- ²⁵M. Gallegos, V. Vassilev-Galindo, I. Poltavsky, A. Martín Pendás, and A. Tkatchenko, “Explainable chemical artificial intelligence from accurate machine learning of real-space chemical descriptors,” *Nature Communications* **15**, 4345 (2024).
- ²⁶A. R. Oganov and M. Valle, “How to quantify energy landscapes of solids,” *The Journal of Chemical Physics* **130**, 104504 (2009).
- ²⁷M. Karabin and D. Perez, “An entropy-maximization approach to automated training set generation for interatomic potentials,” *The Journal of Chemical Physics* **153** (2020).
- ²⁸D. M. de Oca Zapiain, M. A. Wood, N. Lubbers, C. Z. Pereyra, A. P. Thompson, and D. Perez, “Training data selection for accuracy and transferability of interatomic potentials,” *npj Computational Materials* **8** (2022), 10.1038/s41524-022-00872-x.
- ²⁹B. W. Silverman, *Density Estimation for Statistics and Data Analysis* (Taylor & Francis, 1986).
- ³⁰D. Schwalbe-Koda, D. E. Widdowson, T. A. Pham, and V. A. Kurlin, “Inorganic synthesis-structure maps in zeolites with machine learning and crystallographic distances,” *Digital Discovery* **2**, 1911–1924 (2023).
- ³¹L. Himanen, M. O. Jäger, E. V. Morooka, F. F. Canova, Y. S. Ranawat, D. Z. Gao, P. Rinke, and A. S. Foster, “DScribe: Library of descriptors for machine learning in materials science,” *Computer Physics Communications* **247**, 106949 (2020).

AUTHORS' RESPONSE TO THE REVIEWERS

We thank the reviewers for their time and assessment of our work. Since the last manuscript version, we improved the manuscript to address the comments from the reviewers, including clarifying the main objectives of the work and other statements. Below, we provide a point-by-point response to the comments. The reviewer comments are in blue text, while our responses are written in black text.

REVIEWER 1

Reviewer: The new version of the manuscript has a sharper focus which greatly improves the presentation. It makes a convincing case that information entropy is a tool that should be included in the atomistic toolkit and that it offers promising characteristics. As such, the paper could be a useful contribution to the literature. However, I believe that authors still overstate some points, especially those that attempt at quantitative statements, and that these should be tempered before publication.

Authors: We thank the reviewer for the positive evaluation of our manuscript. In this response letter, we clarify the remaining points raised by the reviewer and address the concerns about our statements.

Reviewer: For example, in Section 2.3 the authors write that “Rather, it provides a conservative, yet deterministic way to compute uncertainties by calculating the relative information content of a data point using the training set as a reference, while making no assumptions on the extrapolation power of the model that was trained on these data points. Therefore, our UQ method can be useful to provide rigorous bounds to epistemic uncertainty regimes,” As far as I can tell, no method to predict uncertainties or confidence intervals is actually presented. Rather, an empirical correlation of an overlap measure between test and train sets and force errors is given. While this is an interesting observation, this falls short of a quantification of errors, at best is it a “qualification of errors” (can the test errors be expected to be large or small?). I doubt that a true predictive algorithm would be possible, given the “model-free” nature of the approach. Certainly, different ML models trained and tested to the same data would perform differently, which the approach presented here has no way of capturing.

Authors: We agree with the reviewer that our statement was unclear, and clarified this in the revised version of the manuscript. To continue the discussion proposed by the reviewer, below we address the depth of the question in a point-by-point basis.

Model-free assumption: we agree with the reviewer that our method is an estimate that makes a distinct assumption with respect to existing UQ methods in the literature, namely asking the question “*what if we estimated an uncertainty by decoupling uncertainty due to data from the uncertainty due to model behavior?*” This simplification, as the reviewer pointed out, is unable to estimate errors due to model prediction. If we were concerned mostly with aleatoric uncertainty, this assumption would not make sense. However, it is well-known that epistemic uncertainty is often the root of many problems for machine learning interatomic potentials (MLIPs), which is why we focused on this latter uncertainty type. We showed in our manuscript that our assumption leads to excellent correlations with actual model errors even for state-of-the-art models known to exhibit good behavior even in generalization conditions. The reviewer’s question on whether this could lead to a true predictive UQ is interesting, because we believe that decoupling of data and model uncertainty is one possible route to create a predictive UQ that estimates uncertainties from generalization and model prediction separately. As we also discussed in our previous letter, our assumption is that higher information gain with respect to the training set will lead to higher error regardless of the model, and the error will be larger as the distance becomes larger. We stated this as an “upper bound”, as a good model is likely to reduce this uncertainty and, ideally, make it vanish by being perfectly able to extrapolate.

UQ vs. empirical correlations: Regarding the issue of empirical correlations as UQ method, one can argue that, as of today, most, if not all, UQ methods for atomistic simulations are empirical correlations between prediction errors and a surrogate quantity. For instance, one method that is somewhat closer to ours was proposed by Zhu et al.,¹ where they fit a Gaussian mixture model (GMM) to the distribution of latent representations of a neural network (NN) model. In that case, the quantity that relates to uncertainty is the negative log-likelihood (NLL) of a sample with respect to the (parametric) GMM. In essence, the NLL does not follow the units of the actual quantity to be predicted (i.e., force errors), but is demonstrated to have good correlation with the actual errors of the model prediction from this empirical demonstration. Another classical example is the query-by-committee approach, which relies on the fact that NNs trained on different subsets of data, or even

just initialized/trained with different random seeds, exhibit different extrapolation behavior outside of the training domain. In these examples, the uncertainty is approximated by the variance of model predictions. Whereas this method is known to work well in practice, there are no guarantees that the predictions outside of the training domain will be different, and the method may only be more reliable as the number of NNs in the committee increases. Numerous recent papers have been studying these correlations, and we recommend the work of Tan et al.² as a good comparison of these different strategies. Nevertheless, we point out that the correlation between errors and uncertainty metric is the way the field has been reporting these findings — even if is an incomplete picture of the ideal UQ.

Quantification vs. “qualification” of uncertainty with $\delta\mathcal{H}$: To address one of the reviewer’s concerns regarding the mismatch between our uncertainty measure $\delta\mathcal{H}$ and the actual error, we used the data shown in Fig. 3d of the manuscript and performed a conformal prediction to map between our surrogate uncertainty measure $\delta\mathcal{H}$ and actual machine learning errors. This conformal prediction approach uses a subset of the data and performs a quantile regression to map between the two distributions and has been used extensively in other works within MLIPs.^{2,3} This model is later used to estimate the error of the rest of the distribution, thus effectively performing an error quantification. These results are shown in Fig. R1. The predicted error is computed for the 50% of the dataset that are not used to fit the conformal prediction model. Figure R1 shows that performing a map from $\delta\mathcal{H}$ to the error is fairly trivial with a conformal prediction. The correlation between the predicted and true error values is aligned with other results in the field^{1,2} and — despite not making any model about the MLIPs — exemplify how a more complete UQ can be performed with our approach.

UQ with $\delta\mathcal{H}$ vs. dataset overlap: We also point out that, contrary to the reviewers’ comment, the measures of overlap and differential entropy are different. The overlap between two datasets $\{\mathbf{X}\}$ and $\{\mathbf{Y}\}$ is computed as the fraction of values $\delta\mathcal{H}(\mathbf{Y}_i|\{\mathbf{X}\}) \leq 0, \forall \mathbf{Y}_i \in \{\mathbf{Y}\}$, and is thus a measure defined for a *dataset given another dataset*. The uncertainty of an environment \mathbf{Y}_i given a reference dataset $\{\mathbf{X}\}$ is the actual value of $\delta\mathcal{H}(\mathbf{Y}_i|\{\mathbf{X}\})$, thus representing a point-wise quantity that does not average out the dataset results, and is a measure defined for an *environment with respect to a dataset*. This is how Fig. 3c of our manuscript (also reproduced in Fig. R1 after the conformal prediction) is created. It uses the distribution of point-wise errors and $\delta\mathcal{H}$ values rather than the overlap. These

FIG. R1. Relationship between predicted errors and actual errors of a MACE model trained to the “Defects” subset of the GAP-20 dataset. The predicted errors are obtained using a conformal prediction of the values of $\delta\mathcal{H}(\mathbf{X} \mid \text{Defects})$ for half of the dataset, and tested (and shown) for the other half of the dataset. The values of r are the Pearson correlation coefficients for the log of the errors. The orange line showcases the perfect prediction $y = x$.

quantities are shown to exhibit very reasonable correlations for the example in GAP-20, especially when one compares against other model-based UQ methods.²

Summary: To summarize, we agree with the reviewer that, strictly speaking, a complete UQ strategy requires more than demonstrating the correlations, and there are multiple works in the field focusing exclusively in creating the adequate statistical models, e.g., see Ref. 4 for a recent work on the topic. As a whole, the field of MLIPs has yet to make the “quantification” aspect of UQ more rigorous, perhaps with more specific standards and benchmarks. Nevertheless, one of the arguments we wanted to make with this work is that separating

model and data uncertainties purposefully understates the generalization performance of the model, though it makes sense from an empirical risk minimization perspective. If we consider that errors are only minimized for points in the training set, then the assumption of higher errors away from the training set is natural, even if incomplete. Good models may be able to extrapolate well — and many researchers in the field are working to improve this. If that is the case, then our “uncertainty descriptor” $\delta\mathcal{H}$ captures what would happen if a model was unable to extrapolate, but able to perfectly memorize the training data. These are the main assumptions, which, even if not perfectly equivalent to the problem at hand, are demonstrated to have useful correlations with the underlying problem.

To address the reviewer’s concern, we removed the comment about “rigorous bounds” and replaced it with a more clear description of the assumptions and the equivalence of the UQ problems. Regarding the rigor of the uncertainty quantification, we added this point to the discussion section, and appreciate the reviewer’s point in bringing this up. We also added Fig. R1 to the Supplementary Information (new Fig. S19) and describe it briefly in the main text, adding more information about it in the Supplementary Methods. However, because entire papers can be written on the true UQ part of the work, we decided not to elaborate extensively on the “quantification” aspects of the uncertainty, error prediction, and conformal prediction beyond the description of the methods. Therefore, we keep this example in the manuscript for completeness to the reader. We hope this reply also clarifies the remaining concerns from the reviewer regarding this point.

Changes to the manuscript:

Sec. 2.3: Differently from other approaches, however, our method performs a fast non-parametric estimate directly on the atomistic data space, ~~thus bypassing the need for a model~~. While this approach can be expensive for large datasets, it is easily parallelizable and ~~is guaranteed to provide a robust uncertainty estimate, as it does not rely on the randomness associated with model training or inference. Rather, it provides a conservative, yet deterministic way to compute uncertainties by calculating the relative information content of a data point using the training set as a reference, while making and deterministic measure that correlates with uncertainties, which are often challenging or expensive to capture within model-based UQ.~~ Our method also

~~makes~~ no assumptions on the ~~extrapolation-generalization~~ power of the model that was trained on these data points. ~~Therefore, our UQ method can be useful to provide rigorous bounds to epistemic uncertainty regimes, where model-based UQ approaches typically struggle, and to complement model-based UQ when a more clear understanding of model behavior is expected.~~ and can be seen as a lower bound of performance where such generalization is not expected, similarly to what is already performed with Gaussian process regression methods. Detaching the model uncertainty from the data uncertainty also allows our framework to be incorporated into a variety of workflows without modifying architectures, loss functions, or increasing the associated cost of training and evaluating the models.

[...]

Furthermore, because the uncertainty threshold $\delta\mathcal{H} > 0$ is guaranteed by the theory to denote extrapolation (Supplementary Text, Section A.4), our ~~UQ metric-uncertainty measure~~ detects points outside of the training domain without the need for additional calibration or empirically fitted parameters. Whereas this measure does not have the same units of error, similar to other density estimation strategies in MLIPs, an actual quantification of error values from the computed $\delta\mathcal{H}$ can be performed with calibration or conformal prediction methods. In this case, error estimates can be obtained directly from the computed values of $\delta\mathcal{H}$ (see Fig. S19 and Supplementary Methods) and are demonstrated to correlate well with the actual errors from the model. Thus, our information theoretical approach provides a robust, model-free alternative to quantifying ~~errors in MLIPs and also uncertainties in MLIPs that~~ can be used beyond NN models and does not impose constraints on the model training or inference.

Supp. Info.: Added Fig. S19 [equivalent to R1 from this letter]

Reviewer: In fact, Fig 5c shows a counterexample where the force errors of SNAP are uncorrelated with the differential entropy. In that respect, section 2.4 is more balanced, as it shows that information entropy can explain trends, without predictively quantifying them.

Authors: Interestingly, the result in Fig. 5c is a **feature** of our method, not a bug! The reviewer is correct in saying that the typical, “easy” UQ scenario is for errors to be higher outside of the training set, but the result in Fig. 5c is not a counterexample to our strategy. The discussion in the main text (as well as above) is that higher errors under

“extrapolation conditions” are often expected, but not guaranteed. In many cases with the SNAP potential, forces are still well-predicted outside of the training dataset, which is a feature for this model. Part of our argument, in that case, is that assessing extrapolation in these models goes beyond errors. The example in Fig. 5c shows that, even if one had access to the ground truth and tried to obtain the outlier environments from the simulation given the errors, this would not be possible. In SNAP, the deviation from the physical behavior does not happen because the force error is high, but may be due to local energy traps that guides the simulation to undesired states. Generally, this is a well-known error in MLIPs, which has been related empirically by many researchers,⁵⁻⁸ and known to happen in SNAP. What we showed is that our entropy approach can detect these unconventional environments *even when force errors alone would be unable to capture them*. Specifically, in our simulation, amorphous-like regions embedded in an BCC matrix that end up causing unphysical strain hardening. The formation of these regions is deemed unphysical because plasticity in typically BCC metals is very well-documented and not known to lead to this behavior, even in large-scale simulations.⁹ However, detecting these phases in the absence of an outlier detection tool required expert analysis of this large trajectory containing over 30M atoms. While this is doable for the example shown in the manuscript, it is not scalable for automated analysis of the “sanity” of multiple million-atom simulations.

Therefore, we would like to clarify that the main point of this result is nuanced, specifically: (1) sometimes errors do not tell the complete picture, especially in MLIPs (see Ref. 7 for a full discussion on this); (2) detecting “unphysical” results in simulations such as the dynamic strains in tantalum is reasonable work for scientists in the field, but relies on manual verification or extensive computational development;⁹ (3) the analysis of $\delta\mathcal{H}$ can offer an alternative to perform such detection at scale. As we discussed in the manuscript, using NN ensembles to obtain an uncertainty for a 30M-atom trajectory is not as simple, but the computation of $\delta\mathcal{H}$ can be quite scalable. Even if we stayed in the realm of “qualitative uncertainty”, as proposed by the reviewer, this result is still impactful for a practitioner in the field.

We further clarified this point in the main text.

Changes to the manuscript:

Sec. 2.5: Therefore, even access to the ground truth forces does not allow the classification of a trajectory as failed within these constraints, and instead would rely on human inspection of the trajectory, which can be subjective and is prone to observer error. ~~We note~~ This example further illustrates how even force errors are insufficient to detect outliers, and that the unexpected behavior demonstrated in this case appeared late in the trajectory ~~and is extremely~~. These issues are already challenging to anticipate without outlier detection methods, and even more in simulations where the ground truth is not accessible (e.g., training sets constructed with DFT). On the other hand, our differential entropy detects these outliers without the need for a calibrated threshold, providing a conservative estimate for understanding extrapolation in a model-free approach, and providing “early warning” that can be used for model augmentation, re-evaluation, or retraining.

Reviewer: The conclusion states that “this work proposes a rigorous way to optimize their training process, automate evaluation of information contents in datasets, and assess the performance of the models against well-defined theoretical bounds.” Again, the paper clearly shows indicative correlations between information entropy and the behavior of ML potentials that are very helpful to rationalize trends, but the paper does not propose rigorous algorithms to “optimize the training process” or assess their performance “against well-defined theoretical bounds”. In that respect, I find that the abstract and introduction reflect the contributions of the paper more appropriately.

Authors: We removed this comment from the conclusion and made it reflect the text written in the introduction and abstract. We believe the rewritten introduction, as also proposed by Reviewer 3, better reflect the contributions of the manuscript, including the diversity of different, yet related tasks that can be addressed in atomistic simulations with the framework of information theory.

Changes to the manuscript:

Introduction: In this work, we propose a method to quantify information contents in atomistic simulation data. ~~Building on the concepts~~, and demonstrate its application in a range of tasks related to atomistic simulations and MLIPs. Specifically, building on

the mathematical formalism of information theory, we show that the information entropy from a distribution of atom-centered representations can be used to: (1) explain trends in MLIP errors, ~~even in the absence of model predictions;~~ (2) rationalize dataset analysis/compression, provides a robust UQ, enabling the quantification of diversity, data efficiency, and convergence in active learning loops; (3) provide a model-free uncertainty estimate for ML-driven simulations, ~~and;~~ and (4) detect outliers in atomistic simulations, which can be used ~~for outlier detection in a number of applications. In contrast with other works connecting information entropy and atomistic data~~ to identify failures in ML-driven simulations or even rare events such as the onset of nucleation. Importantly, this work ~~proposes a framework to match~~ goes beyond qualitative parallels with information theory and develops an approach where information contents can be quantified under the assumptions of ~~information theory, thus inheriting its useful properties in that theory.~~ We demonstrate that this allows us to use known results in information theory to enable data compressibility, efficiency, error detection, and others. ~~Using this formalism, we show how information contents can be used to extract physical insights from atomistic datasets, including correlations in error metrics for datasets or explaining outliers in trajectories, while avoiding the need for a trained model~~ many others, in the context of atomistic simulations. This work provides a new tool for atomistic simulations, MLIP development, and UQ for computational materials science, and can be extended to enable faster and more accurate materials modeling beyond predictions of potential energy surfaces (PES).

Conclusions: As increasingly accurate and scalable ML models are proposed for atomistic simulations, this work ~~proposes a rigorous way to optimize their training process, automate evaluation of~~ provides a toolkit that can be used to improve these efforts in several thrusts, including evaluating information contents in datasets, ~~and assess the performance of the models against well-defined theoretical bounds~~ assessing the convergence of active learning loops, guiding the curation of more diverse datasets, and detect anomalies in ML-driven simulations.

Reviewer: Minor point: Similarly, in Section 2.5, the comparison of the critical size from the MD with CNT predictions is a stretch, as it uses a mixture of physical quantities from the potential and from experiments. Compounding these approximations with that inherent

to CNT, it is unclear that this direct comparison really brings any credence to the method in comparison with a more conventional CNA-based nuclei identification.

Authors: The evidence shown in our manuscript and previous rebuttal letter are at odds with the reviewer on this point. In our previous letter, we showed how CNA mostly predicts maximum cluster sizes of *one atom* prior or at the onset of nucleation, in disagreement with the reviewer’s comment. We agree that our results make extreme assumptions, but, in contrast, CNA renders any correlation with the classical nucleation theory (CNT) even worse. The previous results illustrated that conventional methods are completely unable to even estimate the number or size of clusters in a liquid trajectory such as ours, or even “clusters” larger than one atom. Some of us have shown that CNA can only work well by removing the “thermal noise” in the trajectory using other deep learning models beyond the scope of this work.¹⁰

In our estimates of critical nuclei sizes with the CNT, we used the values of melting enthalpy and temperatures obtained from simulations for the Cu potential from Mishin et al.¹¹ and computed by Mendeleev et al.¹² As described in the methods, these quantities allow a more fair comparison with the simulation trajectories that we are encountering. In the discussion, we also mention that the interfacial energies have been obtained from experimental measurements and range between 0.177 and 0.221 J/m².^{13–15} This is why we showed a range of critical nuclei sizes in the Fig. 5f in the previous version of the manuscript.

To address this mismatch between experimental and simulated data, we recomputed the figure and rewrote the discussion to use only parameters determined from the potential. Specifically, the solid-liquid interfacial energy for the Cu potential from Mishin et al.¹¹ was determined by some of us to be 0.177 J/m²,¹⁶ thus agreeing exactly with the experimental data and the results shown in the manuscript. This means that the estimated critical nucleus size, using the computational data, is equal to the smaller end of the range shown in the previous Fig. 5f, which is the best agreement with the simulation results. Therefore, the results remain strong when the results are the ones obtained from simulations, while this cannot be said for the outcomes from CNA, which detects the solid phase afterwards, but not the nucleation event itself.

To clarify this point, we improved the explanation in the manuscript regarding these values, and add the reference towards the computational results for the critical nuclei sizes in Cu. We also removed the range of predictions in Fig. 5f and replaced it by the single

line for the critical nucleus size estimated using the values of interfacial energy, melting temperature, and melting enthalpy for the potential, as well as the average undercooling for each time step.

Changes to the manuscript:

Fig. 5: **d.** Visualization of the solidification trajectory during the nucleation, growth, and solid states. FCC, HCP, and BCC phases are shown with green, red, and blue colors, respectively. Non-identified phases are represented in gray. **e.** Number of FCC atoms derived from the MD simulation with the a-CNA method. The shaded blue area indicates the time window where crystal growth is observed. The critical nucleus is observed around 917 K. The black dots indicate the frames corresponding to nucleation, growth, and final solidified system visualized in **a.** **f.** Largest cluster size in the simulation box, obtained by grouping atoms with $\delta\mathcal{H} < 0$. The orange line represents the estimated critical nuclei sizes estimated using the values of interfacial energy, melting temperature, and melting enthalpy for the potential, and the average undercooling for each time step.

Sec. 2.5: In contrast, the CNA method recovers a maximum of 170 FCC-like environments within the entire simulation box and across all pre-nucleation frames ~~—~~(Fig. S27). To compare this with predictions from the CNT, we calculated the critical nucleus size given the average, time-dependent undercooling in the simulation. The melting enthalpy and temperature obtained from the simulation’s EAM potential were used to perform such estimate, ~~as calculated by Mendeleev *et al.*~~The solid-liquid interfacial energy was ~~taken from~~ also computed for the potential by Zepeda-Ruiz *et al.* and determined to be equal to 0.177 J/m². This value is in excellent agreement with experiments, which range between 0.177 and 0.221 J/m² for Cu. ~~This range of predictions~~The critical nucleus size estimated at each timestep (given the undercooling) and the simulated values for the Cu potential is shown in Fig. 5f in orange color. The results demonstrate that nucleation happens when the largest cluster identified by our information theoretical method falls ~~roughly within the range of experimental critical nucleus sizes~~above the ~~threshold of minimum cluster size for nucleation~~. Prior to the nucleation event, only a single other frame intersects the ~~region of maximum cluster size~~threshold of minimum

cluster size for nucleation. Visualization of the cluster indicates that the graph at that frame is better approximated as two nuclei rather than a single, compact critical nucleus (Fig. 25, see also Sec. A.10). ~~This~~, which may be an artifact of the graph-theoretical approach used to identify the clusters. Nevertheless, this demonstrates that our ~~method~~ information-theoretical approach can be used to ~~reliably and sensitively~~ detect rare events in atomistic simulations, including cases where ground-truth values (e.g., force errors) or existing classification methods (e.g., a-CNA) lack sensitivity to predict.

Reviewer: The code is provided with examples of use and easy means of installation. The functionalities are straightforward and the level of detail in the readme is appropriate.

Authors: We thank the reviewer for their attention to our code and positive evaluation of the methods.

REVIEWER 3

Reviewer: The revisions and replies of the authors have not changed my initial evaluation of the manuscript. Although the new version is more focused, I still don't know what the primary purpose of the article is. From the (at times quite wordy) replies, I understand that they justify their particular choices of the entropy estimation, the descriptors, etc a posteriori by the apparent "utility" of the method for multiple applications in atomistic simulations. This is fine, but it narrows the scope and, in my opinion, makes this work less interesting for researchers outside of the addressed part of the MLIPs community.

Authors: We thank the reviewer once again for the evaluation of the manuscript, and regret that the main points did not come across easily to the reviewer. In the current and previous response letters, we tried to offer a complete discussion with the reviewers and make sure that all concerns are properly addressed. We believe we addressed the original discussions on the limitations of the work, including an extensive description of them in the Supplementary Information and in the main text, and complied with the reviewers' requests for a narrower focus.

Regarding the utility, the field of machine learning potentials is a rapidly expanding community, and is becoming widely used across the entire atomistic simulation community.

Recent publishing trends show that nearly one third of all materials science papers included some flavor of ML in 2023, surpassing over 50,000 papers in that year.¹⁷ The number may be even higher for 2024, and was reported to be around 42% at the time of that publication.¹⁷ Other recent publications show that the process of training and evaluating potentials for multiple applications is quite impactful and can influence the broader atomistic modeling community in powerful ways.^{18–20} Nevertheless, this process of training, evaluating, auditing, and debugging models remains quite empirical and reliant on strategies related to the models themselves. In this context, our work attempts to formalize a range of issues regarding dataset diversity, correlations between errors and data, predicting uncertainties, detecting outliers, and so on. In our revised manuscript, we attempted to summarize these challenges with Fig. 1a of the main text, which showed in magenta color all the challenges that our method addresses.

For example, we are unaware of any work that **quantifies** the diversity of a dataset for MLIPs beyond qualitative dimensionality reduction techniques and describes, with the language of information theory, how much information is contained in a dataset for atomistic simulations. We showed how this quantification enables the computation of efficiency in active learning loops, sampling strategies, and even explains learning curves in some datasets. In several examples, these have been attributed to model performance alone, but we demonstrate the origin of many of these behaviors and, importantly, quantify them with well-defined bounds and an underlying theory.

As another example, all practitioners in the field know that most of the time spent training MLIPs is dedicated to curating the right data and making sure it is representative of the phase space to be explored in the future. In fact, the field has thrived by proposing multiple strategies on how to sample more diverse datasets. Nevertheless, diversity is a measure of datasets, not models, but the field uses surrogate metrics such as model errors to assess diversity. We argue that this can be resolved by removing the model from the equation and analyzing the data with the right toolkit that assesses information contents: information theory. This assumption is completely new to the field of MLIPs, and unlocks a range of explanations previously unavailable.

In yet another demonstration of this, in our revised manuscript, we introduced an entire new section to showcase how we can rationalize error trends in previously unexplained results. The paper describing the TM23 dataset recently proposed by Owen et al. in 2024

contains a number of interesting, open questions about why the results are the way they are. In our work, we showed how many of these open questions can be answered with a simple (yet previously unknown) computation of the dataset entropies, diversities, and uncertainties while removing the dependence on the model.

Furthermore, the narrower scope of the revised manuscript was a direct result of addressing previous comments by this and other reviewers. The present comment that the narrower scope is no longer interesting directly contradicts those previous comments. In our previous letter, we also provided over 10 references of works in this journal that use and/or develop strategies for fitting MLIPs, sometimes for specific (yet important) composition spaces, most of which have demonstrated to be extremely popular examples or influential strategies for the community of atomistic simulations. Other recent works have shown that developing MLIPs is of high interest to the atomistic modeling community as a whole, with several examples of method-enabling works showcasing broad impact within the field.^{18–21} The fact that we demonstrate how our work goes beyond MLIPs and can be used to analyze trajectories, datasets, and so on is a clear demonstration of the utility and novelty of the method beyond plain MLIP fitting, as showcased in Sec. 2.5 and the extensive Supplementary Information of the manuscript. Therefore, we believe that our work remains highly relevant and appropriate for the scope of Nature Communications.

To avoid extending this letter and re-enumerate the innovations of the method already described in the manuscript, we refer the reviewer to our revisions of the Introduction, where we have added an explicit list of these contributions, mirroring an existing list in the conclusion.

Changes to the manuscript:

Introduction: In this work, we propose a method to quantify information contents in atomistic simulation data. ~~Building on the concepts~~, and demonstrate its application in a range of tasks related to atomistic simulations and MLIPs. Specifically, building on the mathematical formalism of information theory, we show that the information entropy from a distribution of atom-centered representations can be used to: (1) explain trends in MLIP errors ; ~~even in the absence of model predictions;~~ (2) rationalize dataset analysis ~~/compression, provides a robust UQ~~, enabling the quantification of diversity, data

efficiency, and convergence in active learning loops; (3) provide a model-free uncertainty estimate for ML-driven simulations, ~~and~~; and (4) detect outliers in atomistic simulations, ~~which~~ can be used ~~for outlier detection in a number of applications. In contrast with other works connecting information entropy and atomistic data~~ to identify failures in ML-driven simulations or even rare events such as the onset of nucleation. Importantly, this work ~~proposes a framework to match~~ goes beyond qualitative parallels with information theory and develops an approach where information contents can be quantified under the assumptions of ~~information theory, thus inheriting its useful properties in that theory. We demonstrate that this allows us to use known results in information theory to enable data compressibility, efficiency, error detection, and others. Using this formalism, we show how information contents can be used to extract physical insights from atomistic datasets, including correlations in error metrics for datasets or explaining outliers in trajectories, while avoiding the need for a trained model~~ many others, in the context of atomistic simulations. This work provides a new tool for atomistic simulations, MLIP development, and UQ for computational materials science, and can be extended to enable faster and more accurate materials modeling beyond predictions of potential energy surfaces (PES).

Conclusions: As increasingly accurate and scalable ML models are proposed for atomistic simulations, this work ~~proposes a rigorous way to optimize their training process, automate evaluation of~~ provides a toolkit that can be used to improve these efforts in several thrusts, including evaluating information contents in datasets, ~~and assess the performance of the models against well-defined theoretical bounds~~ assessing the convergence of active learning loops, guiding the curation of more diverse datasets, and detect anomalies in ML-driven simulations.

Reviewer: In any case, I think that the limitations resulting from some of their choices are not adequately discussed, and some statements are overly broad. For example, in the introduction, they write, “In contrast with other works connecting information entropy and atomistic data, this work proposes a framework to match the assumptions of information theory, ...” (no references given) To be clear: the entropy estimate (Eq. (5)) is well established in information theory, the similarity kernels of atomistic environments have been used and discussed in many works and have been used to calculate information entropy (even if

there might be limitations). If the purpose of the article is to show that their implementation of Eq. (5) (and its ingredients) is useful for “model-free quantification . . .”, they should write that. If they claim to have a new framework (or theory), they should clearly describe what is new and what is known.

Authors: As we discussed in the previous response letter and above, we agree that we are not the first ones to use information entropy in the context of materials science or chemistry. In the Introduction, we properly cited multiple of such works, including the ones previously suggested by the Reviewer. In the revised manuscript, we rephrased the sentence highlighted by the reviewer to improve its clarity and point to our contributions within model-free quantification of completeness, uncertainties, and outliers in atomistic ML. After the reviewer’s suggestion in the previous letter, we had changed the claims of “new theory” already in the previous resubmitted version of the manuscript, and believe it no longer applies to this work with improved focus.

Nevertheless, the Supplementary Information explains in detail the exact issue raised by the reviewer, including what are the limitation of other methods and how our approach addresses them (Supp. Text, Sections A.2 – A.4), including issues raised in the previous letter. Unfortunately, due to space limitations, our manuscript cannot address all of the issues in the main text, but we tried to be as complete as possible in the derivations and discussions in the Supplementary Information. We believe that the answer to this and many other of the points from the reviewer are already contained in that supplementary document.

To address this concern, we added a clarifying point in the main text regarding the use of Eq. (5) in our methodology and its utility in numerous applications, from model-free quantification of errors, uncertainties, and outlier detection.

Changes to the manuscript:

Sec. 2.1: In our definition, $\mathcal{H} = \log n$ implies $K_h(\mathbf{X}_i, \mathbf{X}_j) = \delta_{ij}$, which is the case when all points are dissimilar from each other. $\mathcal{H} = 0$, on the other hand, implies $K_h(\mathbf{X}_i, \mathbf{X}_j) = 1, \forall i, j$, which represents a degenerate dataset with all points equivalent to each other. ~~We discuss the importance of these and other useful properties of~~ This work explores how this and other properties of Eq. (5) are useful for a variety of applications in atomistic simulations, specifically in the quantification of errors, uncertainties, and

outliers in model-free regimes. A comprehensive discussion of the importance of this mathematical approach for estimating the information entropy of atomistic datasets, especially the properties particular to this mathematical formalism, are described in detail in the ~~information entropy for atomistic simulations in detail at the~~ Supplementary Text.

Reviewer: As I wrote before, I think the article is well-written and the results are interesting. I don't share the optimistic interpretation of the importance of the "framework", but it might be useful for some researchers. The usefulness of the framework depends in my view crucially on the ability to go beyond the current assumptions, something the authors prefer to investigate in a later publication. Overall, I still think that the manuscript in its current form is more suitable for a journal focused on cheminformatics or materials informatics.

Authors: Again, we thank the reviewer for the positive evaluation of our manuscript and for the discussion. Our initial proposed work was already much broader in scope, and the reviewers already raised the issue of length in our manuscript. Along with the descriptive Supplementary Information, we believe we addressed the main topic of the manuscript at length and with rigor, and thanks to the reviewers' comments, with improved focus. The reviewers' previous comments also emphasized that the lack of focus was a drawback of the manuscript. We believe, therefore, that the narrower scope was a positive byproduct of the peer review process, and strengthens it instead of weakening it.

Regarding the usefulness, the manuscript already showcases numerous examples that had previously challenged the community in the development of MLIPs, rationalizing them and scrutinizing the results (and limitations of our assumptions). We not only validated our methods, but propose new explanations to trends in the literature that had been left unaddressed. Since our first submitted manuscript, we added entire new sections to demonstrate the importance and utility of the method, going beyond the initial assumptions and tackling large, new datasets with trends previously unexplained, as shown in Sec. 2.4 with the TM23 dataset.²² This substantial added work improved its scope, provided extensive new evidence to the manuscript. We stand by our work regarding its usefulness and potential impact in the community; this stance is not due to sheer optimism, but rather by the strength of the evidence demonstrated in the manuscript and the underlying theory. Plus, as we mentioned above, the community of MLIPs is not a niche area of research, is adopted by thousands

of researchers, and published routinely in this journal. In fact, roughly since we submitted this article for peer review in this journal for the first time (May 2024), Nature Communications has published 7 articles developing or using machine learning potentials, sometimes (reasonably) focusing on specific systems.²³⁻²⁹

One could also argue that the fact that the most interesting research is yet to come is another sign of an impactful publication, and one that extends beyond its own field. For researchers analyzing datasets in atomistic models, we believe the current work can already provide broad impact for an entire community developing MLIPs and performing atomistic modeling, which is a global and immense community. We are naturally working on them as follow ups to our own work, and believe that augmenting this extensive manuscript with even more information will not strengthen it, but weaken the overall message — as the reviewer also pointed out in the previous letter. Currently, nearly at 8,000 words of main text, 5 figures, 4 pages of Methods, and 44 pages of Supplementary Information — 15 of which are single-spaced text detailing the derivations, assumptions, limitations, and methods — we are already at the limit of what most journals accept, so extending the work without removing extensive parts of this manuscript (which would compromise its quality) is not an ideal strategy. Finally, Nature Communications is a broad journal, but one that also has categories and subcategories for manuscripts, including one that we believe is quite appropriate to our manuscript: “Materials Science/Theory and computation/Atomistic models”.

REFERENCES

- ¹A. Zhu, S. Batzner, A. Musaelian, and B. Kozinsky, “Fast uncertainty estimates in deep learning interatomic potentials,” *The Journal of Chemical Physics* **158** (2023).
- ²A. R. Tan, S. Urata, S. Goldman, J. C. Dietschreit, and R. Gómez-Bombarelli, “Single-model uncertainty quantification in neural network potentials does not consistently outperform model ensembles,” *npj Computational Materials* **9**, 225 (2023).
- ³Y. Hu, J. Musielewicz, Z. W. Ulissi, and A. J. Medford, “Robust and scalable uncertainty estimation with conformal prediction for machine-learned interatomic potentials,” *Machine Learning: Science and Technology* **3**, 045028 (2022).
- ⁴S. Kounouho, R. Dingreville, and J. Guilleminot, “Stochastic symplectic reduced-order modeling for model-form uncertainty quantification in molecular dynamics simulations in

- various statistical ensembles,” *Computer Methods in Applied Mechanics and Engineering* **431**, 117323 (2024).
- ⁵J. S. Smith, B. Nebgen, N. Mathew, J. Chen, N. Lubbers, L. Burakovsky, S. Tretiak, H. A. Nam, T. Germann, S. Fensin, and K. Barros, “Automated discovery of a robust interatomic potential for aluminum,” *Nature Communications* **12**, 1257 (2021).
- ⁶D. Schwalbe-Koda, A. R. Tan, and R. Gómez-Bombarelli, “Differentiable sampling of molecular geometries with uncertainty-based adversarial attacks,” *Nature Communications* **12**, 5104 (2021).
- ⁷X. Fu, Z. Wu, W. Wang, T. Xie, S. Keten, R. Gomez-Bombarelli, and T. Jaakkola, “Forces are not Enough: Benchmark and Critical Evaluation for Machine Learning Force Fields with Molecular Simulations,” *arXiv:2210.07237* (2022), 10.48550/arXiv.2210.07237, arXiv:2210.07237.
- ⁸D. M. de Oca Zapiain, M. A. Wood, N. Lubbers, C. Z. Pereyra, A. P. Thompson, and D. Perez, “Training data selection for accuracy and transferability of interatomic potentials,” *npj Computational Materials* **8** (2022), 10.1038/s41524-022-00872-x.
- ⁹L. A. Zepeda-Ruiz, A. Stukowski, T. Opperstrup, and V. V. Bulatov, “Probing the limits of metal plasticity with molecular dynamics simulations,” *Nature* **550**, 492–495 (2017).
- ¹⁰T. Hsu, B. Sadigh, N. Bertin, C. W. Park, J. Chapman, V. Bulatov, and F. Zhou, “Score-based denoising for atomic structure identification,” *npj Computational Materials* **10**, 155 (2024).
- ¹¹Y. Mishin, M. Mehl, D. Papaconstantopoulos, A. Voter, and J. Kress, “Structural stability and lattice defects in copper: Ab initio, tight-binding, and embedded-atom calculations,” *Physical Review B* **63**, 224106 (2001).
- ¹²M. Mendeleev, M. Rahman, J. Hoyt, and M. Asta, “Molecular-dynamics study of solid–liquid interface migration in fcc metals,” *Modelling and Simulation in Materials Science and Engineering* **18**, 074002 (2010).
- ¹³D. Turnbull, “Formation of Crystal Nuclei in Liquid Metals,” *Journal of Applied Physics* **21**, 1022–1028 (2004).
- ¹⁴B. Vinet, L. Magnusson, H. Fredriksson, and P. J. Desré, “Correlations between Surface and Interface Energies with Respect to Crystal Nucleation,” *Journal of Colloid and Interface Science* **255**, 363–374 (2002).
- ¹⁵G. Kaptay, “A coherent set of model equations for various surface and interface energies

- in systems with liquid and solid metals and alloys,” *Advances in Colloid and Interface Science* **283**, 102212 (2020).
- ¹⁶L. Zepeda-Ruiz, B. Sadigh, A. Chernov, T. Haxhimali, A. Samanta, T. Ooppelstrup, S. Hamel, L. Benedict, and J. Belof, “Extraction of effective solid-liquid interfacial free energies for full 3d solid crystallites from equilibrium md simulations,” *The Journal of chemical physics* **147** (2017).
- ¹⁷K. T. Butler, K. Choudhary, G. Csanyi, A. M. Ganose, S. V. Kalinin, and D. Morgan, “Setting standards for data driven materials science,” *npj Computational Materials* **10**, 231 (2024).
- ¹⁸C. Chen and S. P. Ong, “A universal graph deep learning interatomic potential for the periodic table,” *Nature Computational Science* **2**, 718–728 (2022).
- ¹⁹A. Merchant, S. Batzner, S. S. Schoenholz, M. Aykol, G. Cheon, and E. D. Cubuk, “Scaling deep learning for materials discovery,” *Nature* , 80–85 (2023).
- ²⁰I. Batatia, P. Benner, Y. Chiang, A. M. Elena, D. P. Kovács, J. Riebesell, X. R. Advincula, M. Asta, W. J. Baldwin, N. Bernstein, *et al.*, “A foundation model for atomistic materials chemistry,” arXiv:2401.00096 (2023).
- ²¹S. Batzner, A. Musaelian, L. Sun, M. Geiger, J. P. Mailoa, M. Kornbluth, N. Molinari, T. E. Smidt, and B. Kozinsky, “E(3)-equivariant graph neural networks for data-efficient and accurate interatomic potentials,” *Nature Communications* **13**, 2453 (2022).
- ²²C. J. Owen, S. B. Torrisi, Y. Xie, S. Batzner, K. Bystrom, J. Coulter, A. Musaelian, L. Sun, and B. Kozinsky, “Complexity of many-body interactions in transition metals via machine-learned force fields from the TM23 data set,” *npj Computational Materials* **10**, 92 (2024).
- ²³A. Erlebach, M. Šípka, I. Saha, P. Nachtigall, C. J. Heard, and L. Grajciar, “A reactive neural network framework for water-loaded acidic zeolites,” *Nature Communications* **15**, 4215 (2024).
- ²⁴X. Zhang, S. V. Divinski, and B. Grabowski, “Ab initio machine-learning unveils strong anharmonicity in non-arrhenius self-diffusion of tungsten,” *Nature Communications* **16**, 394 (2025).
- ²⁵K. Song, R. Zhao, J. Liu, Y. Wang, E. Lindgren, Y. Wang, S. Chen, K. Xu, T. Liang, P. Ying, *et al.*, “General-purpose machine-learned potential for 16 elemental metals and their alloys,” *Nature Communications* **15**, 10208 (2024).

- ²⁶S. Roy, J. P. Dürholt, T. S. Asche, F. Zipoli, and R. Gómez-Bombarelli, “Learning a reactive potential for silica-water through uncertainty attribution,” *Nature Communications* **15**, 6030 (2024).
- ²⁷M. Liu, J. Wang, J. Hu, P. Liu, H. Niu, X. Yan, J. Li, H. Yan, B. Yang, Y. Sun, *et al.*, “Layer-by-layer phase transformation in ti3o5 revealed by machine-learning molecular dynamics simulations,” *Nature Communications* **15**, 3079 (2024).
- ²⁸D. Hedman, B. McLean, C. Bichara, S. Maruyama, J. A. Larsson, and F. Ding, “Dynamics of growing carbon nanotube interfaces probed by machine learning-enabled molecular simulations,” *Nature Communications* **15**, 4076 (2024).
- ²⁹D. Fan, S. Naskar, and G. Maurin, “Unconventional mechanical and thermal behaviours of mof calf-20,” *Nature Communications* **15**, 3251 (2024).

AUTHORS' RESPONSE TO THE REVIEWERS

We thank Reviewer 3 one more time for their time and assessment of our work. Below, we provide a point-by-point response to the last comment, as requested by the editorial office. The reviewer comments are in blue text, while our responses are written in black text.

REVIEWER 3

Reviewer: I cannot say that I am convinced by the reply, but the revisions address my comments to some extent. I hope that the authors keep in mind that non-parametric entropy estimation is not unproblematic (especially when kernel density estimates are used) and should be adopted and checked for each data set.

Authors: We thank the reviewer for the time and consideration throughout the discussion and evaluation of this work. We also appreciate the positive feedback throughout the discussion despite the disagreements. We continue to believe that the extensive evidence shown in the manuscript is strong, useful, and novel to warrant a publication in this journal, whereas improvements on the methodology will continue to be performed over time, as is typical in science.

Regarding the non-parametric estimation and the problems with the bandwidth/KDE, the first round of reviews already addressed this point extensively. At the time, we expanded on several parts of the manuscript that touched on this topic. For instance, we have the following excerpt from the main text, Section 2.1:

To employ Eq. (5) in practice, we choose K_h to be a Gaussian kernel,

$$K_h(\mathbf{X}_i, \mathbf{X}_j) = \exp\left(\frac{-\|\mathbf{X}_i - \mathbf{X}_j\|^2}{2h^2}\right),$$

where the bandwidth h is selected to rescale the metric space of \mathbf{X} according to the distance between two FCC environments with a 1% strain (Supplementary Text, Sec. A.6). Nevertheless, as the choice of kernel is known to influence the estimated distribution, the bandwidth (and associated entropy) may vary according to the kernel. Within this work, the bandwidth was kept constant, and

was found to be reasonably adequate for all the tasks that adopt this Gaussian kernel.

Section A.6 from the Supplementary Text described above also contains the following discussion:

The non-parametric estimation of the information entropy H described in Eq. (S13) requires fitting a KDE to the data distribution. In the current work, this selection is challenging due to two issues: (1) differences in density lead to changes in the metric space of the descriptors \mathbf{X} ; and (2) differences in entropy can vary with the choice of bandwidth. To simplify the problem, we selected a bandwidth of 0.015 \AA^{-1} , adopted as constant in this work (except in Sections A.11.2 and A.11.4). As described in the Methods, this corresponds roughly to the distance between two FCC environments ($k = 32$, $r_{\text{cut}} = 5 \text{ \AA}$) with an equilibrium lattice parameter of 3.58 \AA and another with unit cell parameters rescaled by 1% (see Fig. S6). The use of this bandwidth to match different units of entropy is described in Sec. A.11.2.

Therefore, we believe this discussion is already present in the manuscript. To make sure this is even clearer, we emphasized this one more time in Section 2.1 by modifying the sentence above, as described below.

Changes to the manuscript:

Nevertheless, as the choice of kernel is known to influence the estimated distribution, the bandwidth (and associated entropy) may vary according to the kernel and may have to be calibrated depending on the dataset. Within this work, we show that even a constant bandwidth was found to be quite adequate for a range of datasets and tasks adopting this Gaussian kernel.